# Short-wave magnons with multipole spin precession detected in the topological bands of a skyrmion lattice
Ping Che [1,8,9] ✉, Riccardo Ciola [2,9], Markus Garst [2,3] ✉, Volodymyr Kravchuk [2,4], Priya R. Baral [5], Arnaud Magrez [5], Helmuth Berger [5], Thomas Schönenberger [6], Henrik M. Rønnow [6] & Dirk Grundler [1,7] ✉

Topological magnon bands enable uni-directional edge transport without backscattering, enhancing the robustness of magnonic circuits and providing a novel platform for exploring quantum transport phenomena. Magnetic skyrmion lattices, in particular, host a manifold of topological magnon bands with multipole character and non-reciprocal dispersions. These modes have been explored already in the short and long wavelength limit, but previously employed techniques were unable to access intermediate wavelengths comparable to inter-skyrmion distances. Here, we report the detection of such magnons with wavevectors $|q| \simeq 48$ rad μm$^{-1}$ in the metastable skyrmion lattice phase of the bulk chiral magnet $Cu_2OSeO_3$ using Brillouin light scattering microscopy. Thanks to its high sensitivity and broad bandwidth various multipole excitation modes could be resolved over a wide magnetic field regime. Besides the known counterclockwise, breathing and clockwise modes with dipole character, quantitative comparison of frequencies and spectral weights to theoretical predictions enabled the additional identification of a quadrupole mode and, possibly, a sextupole mode. Our work highlights the potential of skyrmionic phases for the design of magnonic devices exploiting topological magnon states at GHz frequencies.

Non-collinear skyrmion spin textures with linear dimensions of tens of nanometers are stabilized in noncentrosymmetric chiral magnets due to the bulk Dzyaloshinskii-Moriya interaction (DMI)[1–9]. Skyrmions possess a topological character and spontaneously condense into regular hexagonal lattices, which makes them appealing for magnonic applications and novel computing[10,11]. These periodic arrangements function as natural magnonic crystals without challenging nano-fabrication and offer the possibility of a bottom-up engineering of magnon band structures in the exchange-dominated spin-wave regime, i.e., at ultrashort wavelength[12,13]. Remarkably, the real-space topology of skyrmion textures is reflected in a non-trivial reciprocal-space topology of the magnon bands, which gives rise to robust magnonic edge states exhibiting unique transport phenomena, such as the magnon Hall effect[13–17].

Via such bands, the magnonic quantum Hall effect comes within experimental reach[18].

The magnonic band structure of skyrmion lattices is characterized by a plethora of excitation modes but their experimental detection is restricted by selection rules. Wave-guide microwave spectroscopy gives access to only the three dipole-active modes close to the $\Gamma$-point of the magnetic Brillouin zone (BZ): the so-called counterclockwise (CCW), breathing and clockwise (CW) modes[19]. These three magnetic resonances have been experimentally investigated in several chiral magnets by microwave spectroscopy[20–25] and other experimental probes like resonant elastic x-ray scattering[26] and time-resolved magneto-optics[27,28]. It was shown recently that the field dependencies of these dipole-active modes are versatile for task-adaptive reservoir computing[11]. Other modes

[1]Laboratory of Nanoscale Magnetic Materials and Magnonics, Institute of Materials (IMX), École Polytechnique Fédérale de Lausanne (EPFL), Lausanne, Switzerland. [2]Institut für Theoretische Festkörperphysik, Karlsruhe Institute of Technology, Karlsruhe, Germany. [3]Institute for Quantum Materials and Technology, Karlsruhe Institute of Technology, Karlsruhe, Germany. [4]Bogolyubov Institute for Theoretical Physics of the National Academy of Sciences of Ukraine, Kyiv, Ukraine. [5]Crystal Growth Facility, Institut de Physique, École Polytechnique Fédérale de Lausanne (EPFL), Lausanne, Switzerland. [6]Laboratory for Quantum Magnetism, Institute of Physics, École Polytechnique Fédérale de Lausanne (EPFL), Lausanne, Switzerland. [7]Institute of Electrical and Micro Engineering (IEM), École Polytechnique Fédérale de Lausanne (EPFL), Lausanne, Switzerland. [8]Present address: Laboratoire Albert Fert, CNRS, Thales, Université Paris-Saclay, Palaiseau, France. [9]These authors contributed equally: Ping Che, Riccardo Ciola. ✉e-mail: ping.che@epfl.ch; markus.garst@kit.edu; dirk.grundler@epfl.ch

of the magnon band structure generically do not possess a macroscopic alternating current (AC) magnetic dipole moment. They do not yield a microwave response and their characteristics as well as functionalities are unexplored. Only for specific values of the magnetic field, the cubic crystalline environment hybridized the CCW and the breathing resonance, respectively, with a sextupole (denoted as sextupole-1 below) and octupole mode such that these otherwise dark modes left a characteristic frequency gap in the microwave response[25]. The direct detection of multipole modes and exploration of their field dependencies remained elusive. The dispersions, i.e., the wavevector, $\mathbf{q}$, dependences of the three magnetic resonances have been investigated for small wave vectors $|\mathbf{q}| \ll k_{SkL}$ by spin-wave spectroscopy in close vicinity of the $\Gamma$-point, where $k_{SkL}$ is the reciprocal lattice vector of the magnetic skyrmion lattice. In the opposite limit of large wavevectors, $|\mathbf{q}| \gg k_{SkL}$, much beyond the first BZ, inelastic neutron scattering was used to explore the convoluted signal of a manifold of spin wave excitations without resolving individual magnon bands[17]. For intermediate wavevectors, $|\mathbf{q}| \sim k_{SkL}$, neutron spin-echo spectroscopy provided first evidence for the dispersion of only the CCW mode[17]. The dispersion of further magnon modes has so far been largely unexplored experimentally. Yet, the intermediate wavevector regime is crucial for any signal processing and computational scheme based on propagating magnons.

Here, we report the spectroscopy of such magnons of the skyrmion lattice with wavevectors on the order $|\mathbf{q}| \sim k_{SkL}$ in the chiral magnet Cu$_2$OSeO$_3$. We use cryogenic Brillouin light scattering (BLS) with an option for high cooling rates of 25 K per minute and thereby explore the skyrmion lattice in its metastable phase[29–32] at 12 K where the spin-wave damping is low[23]. BLS is ideally suited to probe the regime $|\mathbf{q}| \sim k_{SkL}$ over a wide range of frequencies. At such wavevectors, the selection rules of BLS provide a finite spectral weight for multiple modes offering the possibility to detect so far unexplored magnon modes beyond the ones known from waveguide microwave spectroscopy[33]. Particularly, we apply micro-focused BLS using a green laser with wavelength $\lambda_{in} = 532$ nm. The micro-focused setup allows to explore a microscopic volume in the bulk crystal, hence resolving the spin dynamics of the skyrmion lattice in an individual magnetic domain. Stokes and Anti-Stokes spectra are detected corresponding to the emission and absorption of magnons, respectively, with wavevectors $|\mathbf{q}|$ covering values up to 48 rad μm$^{-1}$ within the plane of the skyrmion lattice. Detailed comparisons with a basically parameter-free theoretical prediction for the BLS spectra enable us to identify various excitation modes. Going beyond previous studies, we observe the CCW, breathing and CW modes at finite wavevectors whose frequencies, due to their dispersions, differ substantially from the ones detected close to the $\Gamma$-point. We present experimental evidence for an additional quadrupole mode and, consistent with theory, a weak sextupole mode, denoted, respectively, as quadrupole-2 and sextupole-2 below. In addition, we demonstrate that the breathing mode in the BLS spectrum hybridizes for a specific value of the magnetic field with a decupole mode. Our findings are, on the one hand, key for the fundamental understanding and adequate modeling of magnetic skyrmion lattices as the spin-wave dispersions of the various detected modes depend decisively on symmetric and antisymmetric exchange interactions as well as dipolar coupling between non-collinear spins. On the other hand, our data confirm quantitatively the manifold of minibands providing the basis to explore further the functionalities of topological magnon bands in view of applications such as multi-frequency reservoir computing[11].

## Results

### Experimental setup and skyrmion lattice in Cu$_2$OSeO$_3$

The experimental setup is sketched in Fig. 1a. The BLS laser is focused on the polished top (1$\bar{1}$0) surface of a 300 μm thick Cu$_2$OSeO$_3$ platelet (described in "Materials and Methods" section), and the external magnetic field $\mathbf{H}$ is applied along its [001] crystallographic orientation. The hexagonal skyrmion lattice crystallizes within the (001) plane perpendicular to the applied field and skyrmion tubes extend along $\mathbf{H}$. According to ref. 34, one of the reciprocal lattice vectors of the skyrmion lattice is expected to be aligned with [100] for this field direction, and thus one of its primitive lattice vectors is pointing along [010], see Fig. 1b.

The green laser light with wavelength $\lambda_{in} = 532$ nm is incident along the $-\mathbf{z}$ direction with linear polarization along $\mathbf{y}$ and the back-reflected light propagating along $\mathbf{z}$ is detected with linear polarization along $\mathbf{x}$, where the unit vectors in the crystallographic basis are given by $\mathbf{x}^T = (0, 0, 1)$, $\mathbf{y}^T = -\frac{1}{\sqrt{2}}(1, 1, 0)$ and $\mathbf{z}^T = \frac{1}{\sqrt{2}}(1, -1, 0)$, see Fig. 1a. Before impinging on the sample, the light beam is focused using an optical lens that generates a conical incident-angle distribution with respect to the $z$-axis, i.e., the crystallographic [1$\bar{1}$0] direction, with a cone angle $\theta_{max} = 33°$ corresponding to a numerical aperture of 0.55. The diameter of the focus spot on the sample surface is ~4 μm suggesting that the focal point is located roughly 7.2 μm below the surface. The lens is also rotating the polarization such that the light cone contains rays with different incoming wavevectors $\mathbf{k}_{in}$ as well as different linear polarizations $\mathbf{e}_{in}$, see Fig. 1c.

At the sample surface the light is refracted according to Snell's law

$$\frac{\sin\theta'_{in/out}}{\sin\theta_{in/out}} = \frac{n_1}{n_2} = \frac{|\mathbf{k}_{in/out}|}{|\mathbf{k}'_{in/out}|}, \tag{1}$$

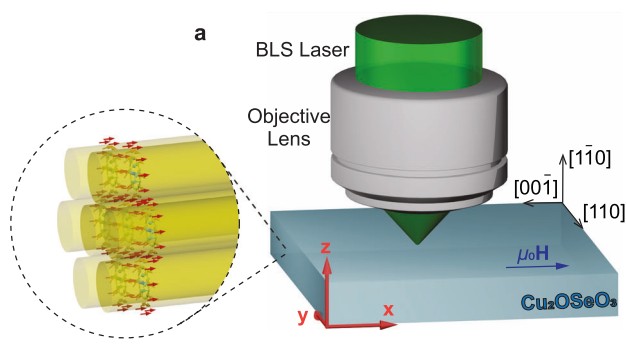

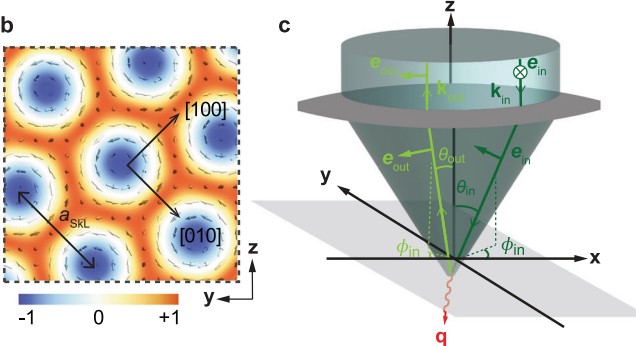

**Fig. 1 | Experimental setup for BLS of magnons in the skyrmion lattice phase of bulk Cu$_2$OSeO$_3$. a** Sketch of the BLS laser focused on the (1$\bar{1}$0) surface of Cu$_2$OSeO$_3$. The external magnetic field $\mathbf{H}$ is applied along [001], i.e., the **x** axis. Skyrmion tubes align with the magnetic field and form a hexagonal lattice within the **y**-**z** plane. **b** Plane of the skyrmion lattice shown in (**a**). The arrows depict the in-plane magnetization and the color coding represent its out-of-plane component. One skyrmion lattice vector connecting two skyrmion centers is assumed to point along [010]. The lattice constant depends weakly on the external magnetic field, and it is on the order of $a_{SkL} \approx 72$ nm. **c** The lens focuses the incoming laser light (dark green) with polarization $\mathbf{e}_{in}$ which leads to a distribution of polarizations that depend on the wavevector of the focused light. The light green arrow represents a photon that is scattered after emitting a magnon (red arrow) in the sample and detected with a polarization filter $\mathbf{e}_{out}$.

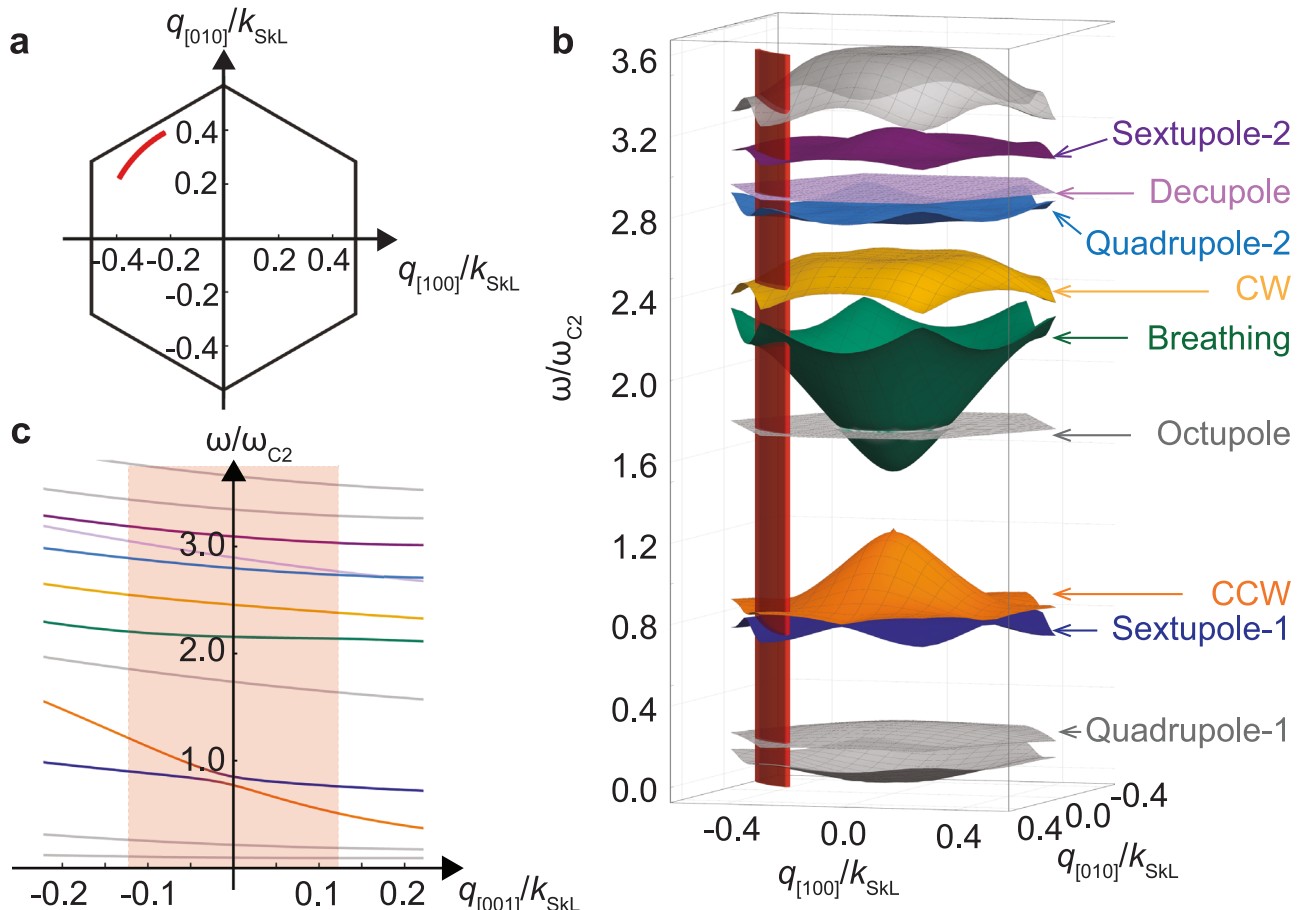

**Fig. 2 | Theoretically calculated magnon band structure of the skyrmion lattice phase for $H = 0.5H_{c2}$. a** First Brillouin zone of the skyrmion lattice with one of the reciprocal lattice vectors, $k_{SkL}$, oriented along $[100]$[34]. **b** Magnon band structure for wavevectors within the plane of the skyrmion lattice. This applies to the case of a scattering wavevector $\mathbf{q} = \mathbf{q}_\perp$ i.e., $\mathbf{q}_\parallel = 0$. The frequency scale $\omega_{c2}$ is defined in the main text below. The $\Gamma$-point is at the center of the reduced zone scheme displayed here. The same coloring of important modes are used throughout this work. **c** Dispersion of the magnon modes as a function of wavevector along the skyrmion tubes at fixed in-plane wavevector $|\mathbf{q}_\perp| = 47$ rad μm$^{-1}$ pointing along $\hat{\mathbf{z}} \parallel [1\bar{1}0]$. The red annulus segments in (**a, b**) as well as the red shaded regime in (**c**) indicate the range of magnon wavevectors accessible with the BLS setup of this work.

where $|\mathbf{k}_{in}| = \frac{2\pi}{\lambda_{in}}$, and the prime identifies angles and wavevectors within the sample. The refractive index outside the material is taken as $n_1 = 1$, while inside $Cu_2OSeO_3$ we assume $n = n_2 \approx 2.03$ at the wavelength of the incoming light $\lambda_{in}$[33]. In particular, this implies $|\mathbf{k}'_{in/out}| = n|\mathbf{k}_{in/out}|$. When the light scatters within the material it transfers momentum $\hbar\mathbf{q}$ and energy $\hbar\omega$ to the sample,

$$\omega = c'(|\mathbf{k}'_{in}| - |\mathbf{k}'_{out}|) = c(|\mathbf{k}_{in}| - |\mathbf{k}_{out}|), \qquad \mathbf{q} = \mathbf{k}'_{in} - \mathbf{k}'_{out}, \qquad (2)$$

where $c' = c/n$ is the speed of light within the material. The Faraday effect was measured in $Cu_2OSeO_3$ at a temperature of 15 K[35], observing a polarization rotation up to 0.003 rad μm$^{-1}$. This effect is four orders of magnitude smaller than the typical momentum transfer considered in the current experiment and will be neglected. The transferred wavevector can be decomposed into two components $\mathbf{q}_\parallel = \hat{\mathbf{x}}q_\parallel$ and $\mathbf{q}_\perp$ that are, respectively, aligned and perpendicular to the applied magnetic field. The component $q_\parallel$ possesses values within the range $\pm 12.9$ rad μm$^{-1}$, and the magnitude of $\mathbf{q}_\perp$ varies from 46.2 rad μm$^{-1}$ to 48 rad μm$^{-1}$. For a more detailed explanation of these wavevector values we refer to the Methods section "theory for the micro-focused BLS cross section".

Importantly, $|\mathbf{q}|$ is on the same order as the reciprocal lattice vector $k_{SkL}$ of the skyrmion lattice in $Cu_2OSeO_3$. The value of $k_{SkL}$ depends weakly on the applied field, but in an extended field range it is approximately given by $k_{SkL}/Q \approx 0.96$ (see Supplementary Fig. 7), with the

wavevector of the helix phase $Q = 105$ rad μm$^{-1}$ in this material. This corresponds to a skyrmion lattice constant of $a_{SkL} = \frac{2}{\sqrt{3}}\frac{2\pi}{k_{SkL}} \approx 72$ nm. The inelastic scattering of such light by emission and absorption of single magnons is thus ideally suited to probe their dispersion close to the edge of the first magnetic BZ. The magnon band structure theoretically expected for the skyrmion lattice phase in cubic chiral magnets[13] is shown in Fig. 2 for $H = 0.5H_{c2}$ with parameters of $Cu_2OSeO_3$, where the critical field $H_{c2}$ borders the field polarized phase existing at large $H$. The hexagonal lattice of skyrmions gives rise to a magnetic BZ sketched in panel a. The circular segment sampled by the component $\mathbf{q}_\perp$ is indicated in red in panel a and b. Panel b displays the dispersion of low-energy spin wave modes of the skyrmion lattice phase for wavevectors $\mathbf{q}_\perp$ within the lattice plane. Colors and labels highlight the modes that are especially important for the present BLS experiment. Only the CCW (orange), breathing (green), and CW (yellow) modes are dipole active at the $\Gamma$-point. Note in particular that due to their strong dispersion the excitation frequency of the CCW and the breathing mode within the experimentally accessible regime of the red segment differ substantially from their frequencies at the $\Gamma$-point. Figure 2c shows the dispersion as a function of $\mathbf{q}_\parallel$ along the skyrmion tubes at a fixed $|\mathbf{q}_\perp| = 47$ rad μm$^{-1}$ pointing along $\hat{\mathbf{z}}$; the red-color-shaded area indicates the range accessible by $\mathbf{q}_\parallel$. The CCW mode possesses a particular strong non-reciprocity as previously discussed in ref. 29.

The other labeled modes in Fig. 2b can be characterized by the angular symmetry of the magnon eigenfunctions that determine the

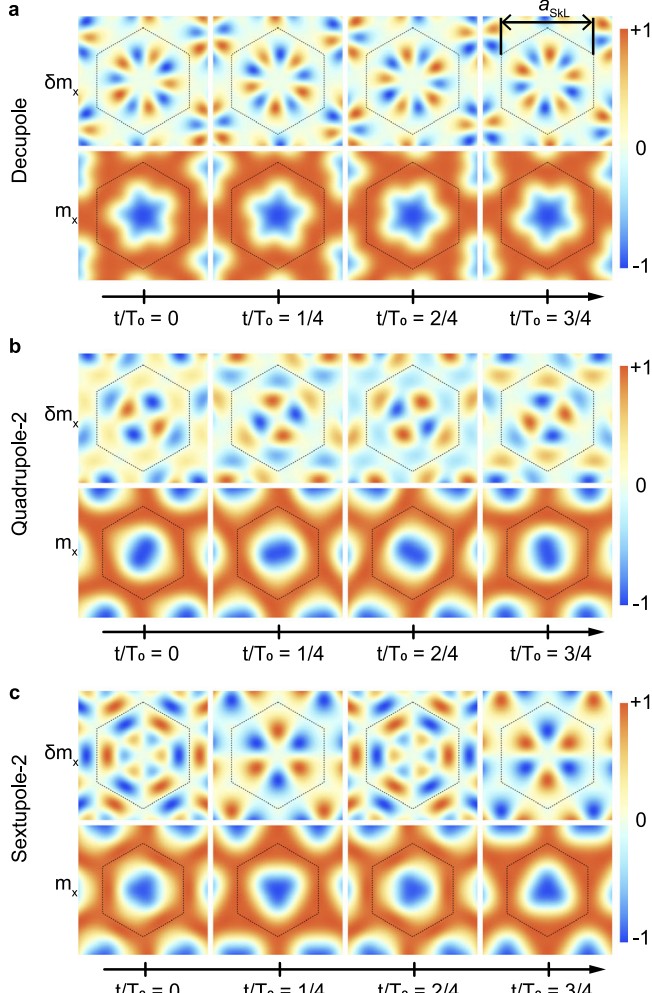

**Fig. 3 | Theoretically calculated spin wave function of the decupole, quadrupole-2 and sextupole-2 modes. a, b, c** illustrate the temporal evolution of the decupole, quadrupole-2 and sextupole-2 modes, respectively, at the $\Gamma$-point of the Brillouin zone where the density plot represents out-of-plane components. The spatial profile of the out-of-plane component $\delta M_x$ of the spin wave functions are shown in the first rows and the resulting deformations of the equilibrium magnetization of Fig. 1b are illustrated in the second rows, where $\mathbf{m} = \mathbf{M}/M_s$ with $M_s$ the saturation magnetization. The hexagon in each panel is the Wigner-Seitz cell whose extension is given by the skyrmion lattice constant that in $Cu_2OSeO_3$ is $a_{SkL} \approx 72$ nm. The corresponding eigenfrequency of each mode, $2\pi/T_0$, is determined by the spectrum of Fig. 2b.

corresponding dynamic deformation of the skyrmion within each unit cell. The sextupole-1 (dark blue) and octupole mode hybridize with the dipole-active modes as reported in ref. 25. The quadrupole-2 (light blue) and sextupole-2 (purple) modes are second-order modes with more involved radial profiles than the respective first-order modes, quadrupole-1 and sextupole-1. The decupole mode (pink) is a first-order mode resulting in a five-fold symmetric deformation of skyrmions within each unit cell.

In Fig. 3 we have illustrated the microscopic nature of the decupole, quadrupole-2 and sextupole-2 modes, respectively, when excited with $q = 0$ at the center of the BZ. The evolution of their spin wave function $\delta \mathbf{M}(\mathbf{r}, t)$ is shown in the first rows of Fig. 3a–c, and it reflects, respectively, the decupolar, quadrupolar and sextupolar character of the three modes. A typical feature of the quadrupole-2 and sextupole-2 modes, that distinguishes them in particular from the corresponding quadrupole-1 and sextupole-1 modes, is the node in their wave function as a function of radial distance for a generic time. This node separates the wave function into an inner ring and an outer ring which rotate as a function of time in opposite directions, see also the

Supplementary Movies (Movie 1 to Movie 5 for CCW, breathing, CW, quadrupole-2 and sextupole-2 modes). The resulting decupole, quadrupole and sextupole deformation of the equilibrium magnetization of Fig. 1b and its time evolution is shown, respectively, in the second row of Fig. 3a–c. The short wavelengths and anti-phase spin-precessional motion inside the Wigner-Seitz cell of the skyrmion lattice make the detection of these modes particularly challenging.

### BLS of magnons of the metastable skyrmion lattice phase

Figure 4a displays the BLS intensity map after a field-cooling (FC) protocol at a rate of 25 K per minute, i.e., the sample was cooled to the temperature of 12 K while applying a finite field $\mu_0 H_{FC} = 16$ mT (red arrow). The process is sketched in Supplementary Fig. 1b. The field $\mu_0 H_{FC}$ was chosen such that during the cooling process the sample passed through the high-temperature skyrmion lattice phase characterized by both the AC susceptibility and BLS spectra conducted at $T = 50$ K (Supplementary Figs. 2 and 3) and realized a metastable skyrmion lattice at 12 K. It has been reported that this metastable stable phase can be extremely robust at low temperature when the magnetic field is applied along cubic axes[31]. When increasing $\mu_0 H$ from 16 mT to about 55 mT, we find BLS spectra with multiple resonances between 1.4 GHz to 7.5 GHz that are clearly distinct from the spectra observed for the conical helix phase, see the "Methods" section on conical and field-polarized (FP) spectra in micro-focused BLS. Both, Stokes at negative frequency and Anti-Stokes signals at positive frequency are plotted corresponding, respectively, to the emission and absorption of spin waves. We find an asymmetry of intensity between the two signals with the Stokes component being enhanced compared to the Anti-Stokes one. At around 55 mT, the spectrum reconstructs in a single branch characterized by linear dispersion. This high-field signature is consistent with the Kittel mode appearing in the FP phase, as explained in the "Methods" section. After warming up the sample to 100 K and cooling down again to 12 K with $\mu_0 H_{FC} = 16$ mT applied, we took spectra for fields smaller than 16 mT. Importantly, the branches at small field align well to the ones detected above 16 mT. We note that selected branches change slopes or fade out for $\mu_0 H \leq 5$ mT indicating a phase transition near zero field. In this work, we focus on fields $\geq 10$ mT.

### Theory for micro-focused BLS of magnons of the skyrmion lattice

The BLS differential cross section for scattering plane waves can be expressed in terms of a correlation function for the fluctuations of the dielectric permittivity[36]

$$\frac{d\sigma_{out,in}(\mathbf{q}, \omega)}{d\Omega'_{out}} \propto \mathbf{e}'_{out,\mu} \mathbf{e}'^{*}_{in,\nu} \mathbf{e}'^{*}_{out,\rho} \mathbf{e}'_{in,\delta} \langle \delta\varepsilon^{*}_{\mu\nu}(\mathbf{r}, t)\delta\varepsilon_{\rho\delta}(\mathbf{r}', t') \rangle_{\mathbf{q}, \omega}, \quad (3)$$

up to a proportionality factor that depends on the frequency of the light. $\mathbf{e}'_{in/out}$ specifies the light polarization within the sample. In the present work, the laser beam is however focused and the incoming light consists of a superposition of plane waves. As explained in detail in the "Methods" section, as the spread of incoming wavevectors $|\Delta \mathbf{k}'_{in}|$ of the focused laser beam is smaller than the reciprocal lattice vectors of the skyrmion lattice, there is no interference between distinct $\mathbf{k}'_{in}$. For our specific setup, we can therefore sum over intensities of in- and out-going wavevectors to obtain the total scattering cross section

$$\sigma(\omega) \propto \int_0^{2\pi} d\phi'_{out} \int_{\pi - \theta'_{max}}^{\pi} d\theta'_{out} \sin\theta'_{out}$$
$$\times \int_0^{2\pi} d\phi'_{in} \int_0^{\theta'_{max}} d\theta'_{in} \sin\theta'_{in} \cos\theta'_{in} \frac{d\sigma_{out,in}(\mathbf{q}, \omega)}{d\Omega'_{out}}. \quad (4)$$

The wavevectors are parametrized according to $\mathbf{k}'_\alpha = |\mathbf{k}'_\alpha|$ $(\mathbf{x} \sin\theta'_\alpha \cos\phi'_\alpha + \mathbf{y} \sin\theta'_\alpha \sin\phi'_\alpha - \mathbf{z} \cos\theta'_\alpha)$, with the index $\alpha$ being *in* or *out*. The sum over out-going wavevectors is such that the scattered photon is reaching the detector. The additional $\cos\theta'_{in}$ factor derives from the

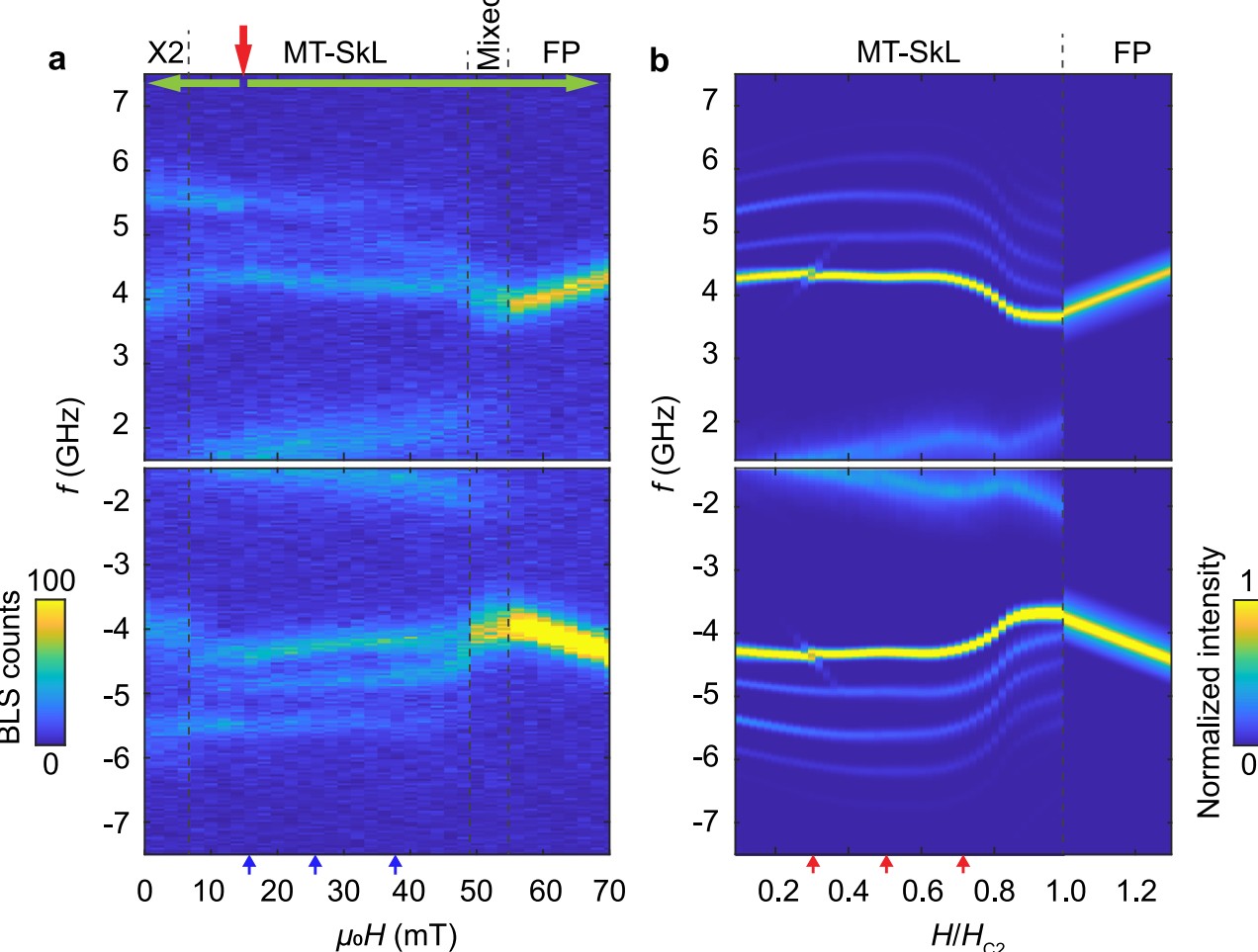

**Fig. 4 | Experimental and theoretical BLS spectra of Cu$_2$OSeO$_3$.** The BLS intensities $I_{BLS}(f) = \sigma(-2\pi f)$ are shown as a function of frequency $f$ where Anti-Stokes, $f > 0$, and Stokes, $f < 0$, processes correspond, respectively, to the absorption and emission of a magnon. **a** BLS intensities for field-cooling at $\mu_0 H_{FC} = 16$ mT (red arrow) down to $T = 12$ K and subsequent field scans (green arrows). Color bar represents the BLS counts. **b** Theoretical BLS spectra of the (metastable) skyrmion lattice phase, $H < H_{c2}$, and the field polarized phase, $H > H_{c2}$, with the same normalization as in (**a**). The entire wavevector domain indicated by the red regions in Fig. 2 contribute to the scattering intensity. Line cuts of the spectra for three values of the magnetic field as indicated by the blue and red arrows at the bottom of (**a** and **b**), respectively, are shown in Fig. 6. We label the metastable skyrmion lattice phase as MT-SkL and field-polarized phase as FP.

homogeneous intensity distribution of the incoming beam. According to $n \sin \theta'_{max} = \sin \theta_{max}$, the cone angle of the focused beam within the sample is $\theta'_{max} \approx 16°$. The BLS intensity displayed in Fig. 4 is related to the scattering cross section via $I_{BLS}(f) = \sigma(-2\pi f)$.

In linear order of spin wave theory, we can expand the magnetization $\mathbf{M}(\mathbf{r}, t) = \mathbf{M}_{eq}(\mathbf{r}) + \delta\mathbf{M}(\mathbf{r}, t)$ up to first order in the spin wave amplitude $\delta\mathbf{M}$ where $\mathbf{M}_{eq}(\mathbf{r})$ is the magnetization profile in equilibrium sketched in Fig. 1b. Magnons will induce fluctuations in the dielectric permittivity,

$$\delta\varepsilon_{\mu\nu}(\mathbf{r}, t) = K_{\mu\nu\lambda}\delta\mathbf{M}_\lambda(\mathbf{r}, t) + 2G_{\mu\nu\lambda\kappa}\mathbf{M}_{eq\lambda}(\mathbf{r})\delta\mathbf{M}_\kappa(\mathbf{r}, t) + \mathcal{O}(\delta M^2), \quad (5)$$

where the tensors $K$ and $G$ comprise the magneto-optic constants for the cubic material. Using the above expression, we can relate the BLS cross section to the dynamical magnetic response function $\chi''_{ij}(\mathbf{q}, \omega)$ attributed to magnons.

The low-energy magnetization dynamics in Cu$_2$OSeO$_3$ is well described by a continuum theory comprising the exchange interaction, DMI, Zeeman term and dipolar interaction. All parameters are known from independent measurements providing a parameter-free prediction of the dynamics. Magnetocrystalline anisotropies are relatively small but are known to stabilize additional phases at low temperatures[31,37–39] and potentially account for the behavior in the small field ranges denoted by $X_2$ and *Mixed* in Fig. 4. As these field ranges are not at the focus of this work, we

neglect magnetocrystalline anisotropies in the following and restrict ourselves to the universal theory valid in the limit of small spin-orbit coupling.

The focused beam generates a distribution of transferred wavevectors $\mathbf{q}$. As a consequence, the transferred energy will cover a range that is determined by the dispersion of the magnon mode, which results in an extrinsic spectral lineshape. In the following theoretical discussion, we limit ourselves to this extrinsic broadening due to the BLS setup using a focused laser beam[40], and we do not consider the small intrinsic damping of spin waves in Cu$_2$OSeO$_3$[23]. The response function $\chi''$ for magnons in the skyrmion lattice phase was previously evaluated in the context of inelastic neutron scattering and we refer the reader to ref. 17 for details. Here, we calculate their BLS intensities. The color-coded intensities are shown in Fig. 4b for $H < H_{c2}$ where various distinct branches can be recognized in the skyrmion lattice phase.

## Quantitative comparison between theoretical and experimental BLS spectra

In order to facilitate the following discussion, the experimental Stokes spectrum is shown again in Fig. 5 overlaid with the theoretical magnon branches calculated for the fixed wave vector $\mathbf{q} = \hat{\mathbf{z}}\, 48\ \text{rad}\,\mu\text{m}^{-1}$ corresponding to an angle of incidence $\theta_{in} = \theta_{out} = 0$ in the present setup. At low absolute frequency most of the spectral weight is attributed to the CCW mode. As magnons with a finite wavevector are probed by BLS a

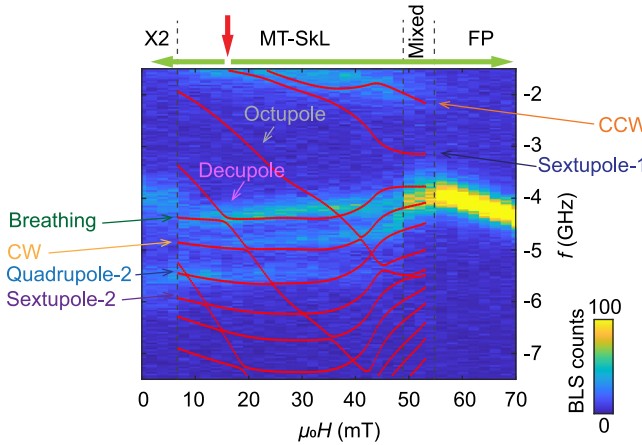

**Fig. 5 | Experimental Stokes spectrum with theoretical magnon dispersions.** Stokes spectrum of Fig. 4a overlaid with magnon frequencies of the metastable skyrmion lattice phase potentially emitted by photons with angle of incidence $\theta_{in} = \theta_{out} = 0$. There is, in particular, experimental spectral weight at higher absolute frequencies that can be attributed to the quadrupole-2 mode, see also Fig. 4b.

hybridization of the CCW with the sextupole-1 mode is expected even in the absence of magnetocrystalline anisotropies[25], see also Fig. 4b. The spectral weight of the measured CCW mode is distributed over a relatively broad frequency range due to its strong dispersion for out-of-plane wavevectors, see Fig. 4a, over which the focused BLS setup collects the scattered photons. The appreciable wavevector distribution and overall small signal-to-noise ratio might explain why the hybridization with the sextupole-1 mode is not resolved experimentally. The narrow branch with relatively large intensity between 4 and 4.5 GHz is ascribed to the breathing mode whose absolute frequency decreases with increasing field in contrast to the CCW mode. The detected difference in eigenfrequencies between the CCW and breathing mode is much larger than that observed with microwave spectroscopy. This large difference reflects the opposite dispersion $f(q)$ in the two minibands with increasing in-plane wavevectors as shown in Fig. 2b. The frequency gap predicted for the avoided crossing with the decupole mode is not resolved. Still, the experimental breathing mode branch exhibits the anticipated small blue shift next to the predicted gap toward small $H$. We attribute the intensity at around 5 GHz to the CW mode. The branch at about 5.5 GHz which stays nearly constant with field $H$ is consistent with the predicted quadrupole-2 mode. Its constant frequency up to about 40 mT is distinctly different from the $+Q$ mode of the previously discussed conical phase whose frequency drops considerably with $H$ (see conical phase spectra in "Methods" section and Supplementary Fig. 4 for comparison). In the theoretical spectra of Fig. 4b the sextupole-2 mode possesses a strikingly small spectral weight. In the experimental intensity map of Fig. 5 this branch is not clearly resolved. We will readdress the sextupole-2 mode when analyzing quantitatively the line cuts shown in Fig. 6. The strongly dispersing octupole mode possesses a negligible BLS spectral weight.

For the quantitative comparison between theory and experiment, we present in Fig. 6 line cuts of the spectral weights at three distinct magnetic fields, 16, 26, and 38 mT, as indicated by the three arrows at the bottom of Fig. 4a, b. The blue symbols are the experimental data of the metastable skyrmion lattice phase, Fig. 4a, and the gray symbols, as a reference, correspond to the conical state, Fig. 7a. The red solid lines are the theoretical BLS spectra. The colored bars at the bottom of each panel identify the various modes which we analyze. The CCW mode (orange) at smallest frequencies possesses a large spectral weight. At 26 mT it hybridizes with the sextupole-1 mode (dark blue) so that these two modes are not clearly separated. The breathing mode (green) gives rise to a large spectral peak positioned between 4 and 4.5 GHz. At 16 mT it hybridizes with the decupole mode (pink), see inset of Fig. 6b. The BLS spectra at positive (Anti-Stokes) and negative (Stokes) frequencies possess features that are reminiscent of this hybridization. The CW mode (yellow) is predicted to exhibit a relatively

small spectral weight. It is better resolved in the Stokes than the Anti-Stokes spectrum. At the finite wavevector probed in the BLS experiment, the quadrupole-2 mode (light blue) is predicted to possess a larger spectral weight than the CW mode, in agreement with the experimental observation. For the sextupole-2 mode (purple), the theoretical spectra show extremely small spectral weights. At 16 mT, the line cut of the experimental BLS data maintains indeed a finite signal strength above the noise floor in the relevant frequency regime of the sextupole-2 mode. With increasing $H$ the predicted signal strength becomes weaker. Correspondingly, the BLS experiment does not resolve the sextupole-2 mode at larger $H$.

In Fig. 6, we depict in light gray color the Anti-Stokes spectra for comparison, which we took in the conical phase by means of the zero-field cooling (ZFC) protocol. There are clear discrepancies in peak positions and field dependencies between the two measurement protocols. The characteristics of modes in the metastable skyrmion lattice phase (blue) are distinctly different from the conical phase (gray).

## Discussion

We evidenced that micro-focus BLS resolves relevant higher order magnon modes of the skyrmion lattice phase which have not been yet addressed by either microwave spectroscopy or neutron scattering previously applied to the chiral magnet Cu$_2$OSeO$_3$. In the backscattering geometry, the BLS probes spin waves with a relatively large wavevector, $q$, on the order of the reciprocal lattice vector $q \sim k_{SkL}$ of the periodic magnetic texture. Thereby, it monitors the dispersion $f(q)$ in an intermediate wavevector regime and is complementary to the other experimental methods that are sensitive to either small, $q \ll k_{SkL}$, or large, $q \gg k_{SkL}$, wavevectors, respectively. This intermediate regime aligns with the typical wavevectors relevant for the development of nanomagnonic circuits.

The good agreement of the experimental data with theoretical predictions for both the eigenfrequencies and the spectral weights has allowed us to identify various modes. We found clear evidence for the wavevector dependent eigenfrequencies of the CCW, breathing and CW modes which were previously studied at the $\Gamma$-point ($q = 0$). In addition, we report on the experimental detection of so far unexplored modes. In particular, a mode with quadrupole character, denoted as quadrupole-2, is well resolved in our spectra over a broad field regime. For a magnon mode with sextupole character, denoted as sextupole-2, small spectral weights are predicted. Its signatures appear with a correspondingly small signal-to-noise ratio in the BLS spectra at 16 mT. At larger $H$, this mode is not resolved experimentally. The multitude of nodes in the spin wavefunction within the Wigner-Seitz cell of Fig. 3b corresponds to pronounced anti-phase precession of neighboring spins belonging to a single skyrmion. At the finite $q$ of the BLS experiment, only a weak net spin precession per skyrmion can remain which explains the challenging detection by inelastic light scattering via the dynamic permittivity tensor components. The mode pattern hence motivates the corresponding small signals of the sextupole-2 mode in both the experimental data and theoretical curves. Similarly, the decupole mode only possesses a small BLS spectral weight for a magnetic field where it hybridizes with the breathing mode, and we find preliminary evidence for such a hybridization in our spectra at 16 mT.

The agreement between the experimental and theoretical BLS spectra, see Figs. 4 and 5, is remarkably good given that the theoretical modeling of the complex micro-focused BLS setup is based on minimal assumptions. We employ the standard theory for cubic chiral magnets valid in the limit of small spin-orbit coupling that neglects magnetocrystalline anisotropies. This theory is characterized by three parameters only[13]: a frequency scale $\omega_{c2} = \gamma \mu_0 H_{c2}^{int}$, where $\gamma = g \mu_B / \hbar$ is the gyromagnetic ratio, $\mu_0$ is the magnetic constant, and $H_{c2}^{int}$ is the internal critical field, a wavevector scale $Q$ and the ratio $\chi_{con}^{int} = M_s / H_{c2}^{int}$, with the saturation magnetization $M_s$, that corresponds to the susceptibility within the conical phase and quantifies the strength of the dipolar interaction. For the latter two parameters we took values from the literature, i.e., $Q = 105$ rad μm$^{-1}$ and $\chi_{con}^{int} = 1.76$ reported in ref. 22. We fitted the frequency scale to our experimental data within the FP phase and obtained $\omega_{c2}/2\pi = 2.03$ GHz that agrees within our error bar with the

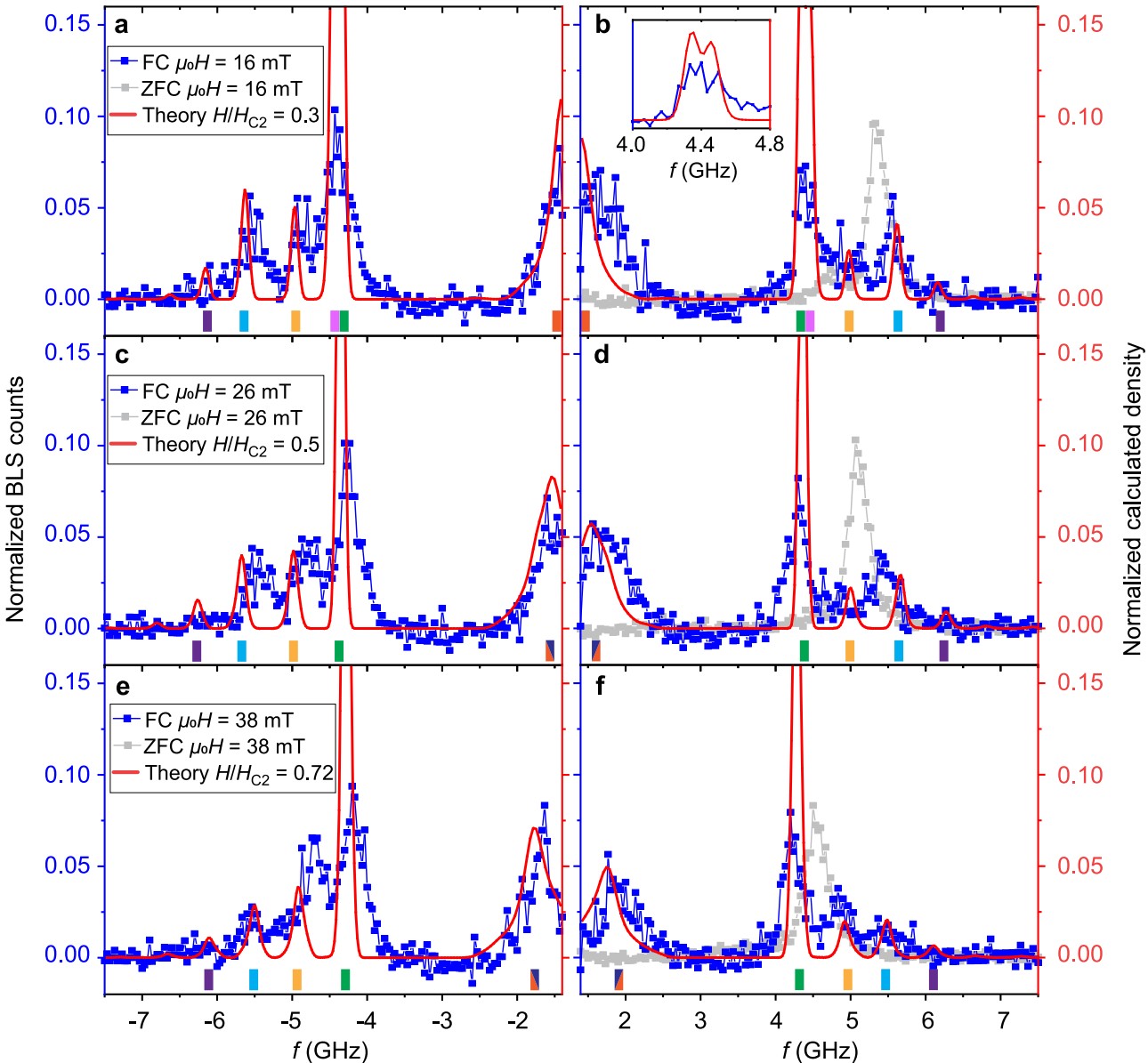

**Fig. 6 | Normalized line cuts of experimental and theoretical BLS spectra.** Line cuts of the spectra in 4a (blue symbols) and Fig. 7a (gray symbols) for three magnetic field values $\mu_0 H$ = 16 mT, 26 mT and 38 mT with Stokes signals shown in (**a, c, e**) and Anti-Stokes signals shown in (**b, d, f**). The blue (gray) symbols are obtained in the metastable skyrmion phase (conical phase). Conical phase intensities have been compressed by a factor of three for direct comparison. The frequency-independent background signal (dark counts) found in the field-polarized phase was removed from the raw BLS spectra to realize a baseline similar to the theoretical curves (red lines). Negative counts in the experimental data (blue symbols) result from the background-signal subtraction. Red solid lines are theoretical BLS spectra for $H/H_{c2}$ = 0.3, 0.5, 0.72. Colored bars at the bottom of each panel indicate positions of theoretical skyrmion lattice resonances using the same color coding as in Fig. 2. Both the experimental and theoretical data have been normalized with respect to the total spectral weight integrated over the full frequency range from −7.5 GHz to −1.4 GHz and from 1.4 GHz to 7.5 GHz. Inset of (**b**) shows the hybridization between the breathing and decupole modes; intensities have been descaled for the direct comparison between the theoretical curve and the experimental data.

value 2.06 found by Ogawa et al.[33]. For the BLS matrix elements we needed, furthermore, the magneto-optic constants in Eq. (5). In a cubic material, the tensor $K_{\mu\nu\lambda} = K\epsilon_{\mu\nu\lambda}$, with the Levi-Civita symbol $\epsilon_{\mu\nu\lambda}$, is characterized by a single constant $K$ that can be absorbed in the overall absolute intensity. The higher-order tensor $G_{\mu\nu\lambda\kappa}$ accounts for the asymmetry of Stokes and Anti-Stokes intensities, and within the FP phase we found a satisfactory fit for the values $G_{11} = G_{12} = 0$ and $2iM_s G_{44}/K \approx 0.123$ in Voigt notation (see Fig. 7c, d). After having fixed all these parameters we obtained, up to the absolute intensity, the parameter-free theoretical prediction for the magnon dispersion and the BLS spectra for the skyrmion lattice phase shown in Figs. 2 and 4, respectively, that describe reasonably well the experimental data.

Corrections beyond the standard theory can be systematically classified in powers of spin-orbit coupling and, already in lowest order, involve many unknown additional coupling constants including magnetocrystalline anisotropies. They are probably relevant to describe the signatures *Mixed* and $X_2$ in the spectra of Fig. 4. In particular, in the low-field regime the metastable skyrmion lattice might undergo an oblique distortion resulting in an elongated skyrmion lattice phase as discussed in ref. 37 and ref. 24. The distortion provides a potential explanation for the reconstruction of the spectra in the field range $X_2$ below 6 mT. Above 50 mT before entering the FP phase in Fig. 4, the spectrum reconstructs with a large intensity close to 4 GHz. In this field range, a conical state, a tilted-conical state or some other magnetic order might be realized resulting in a coexistence of various

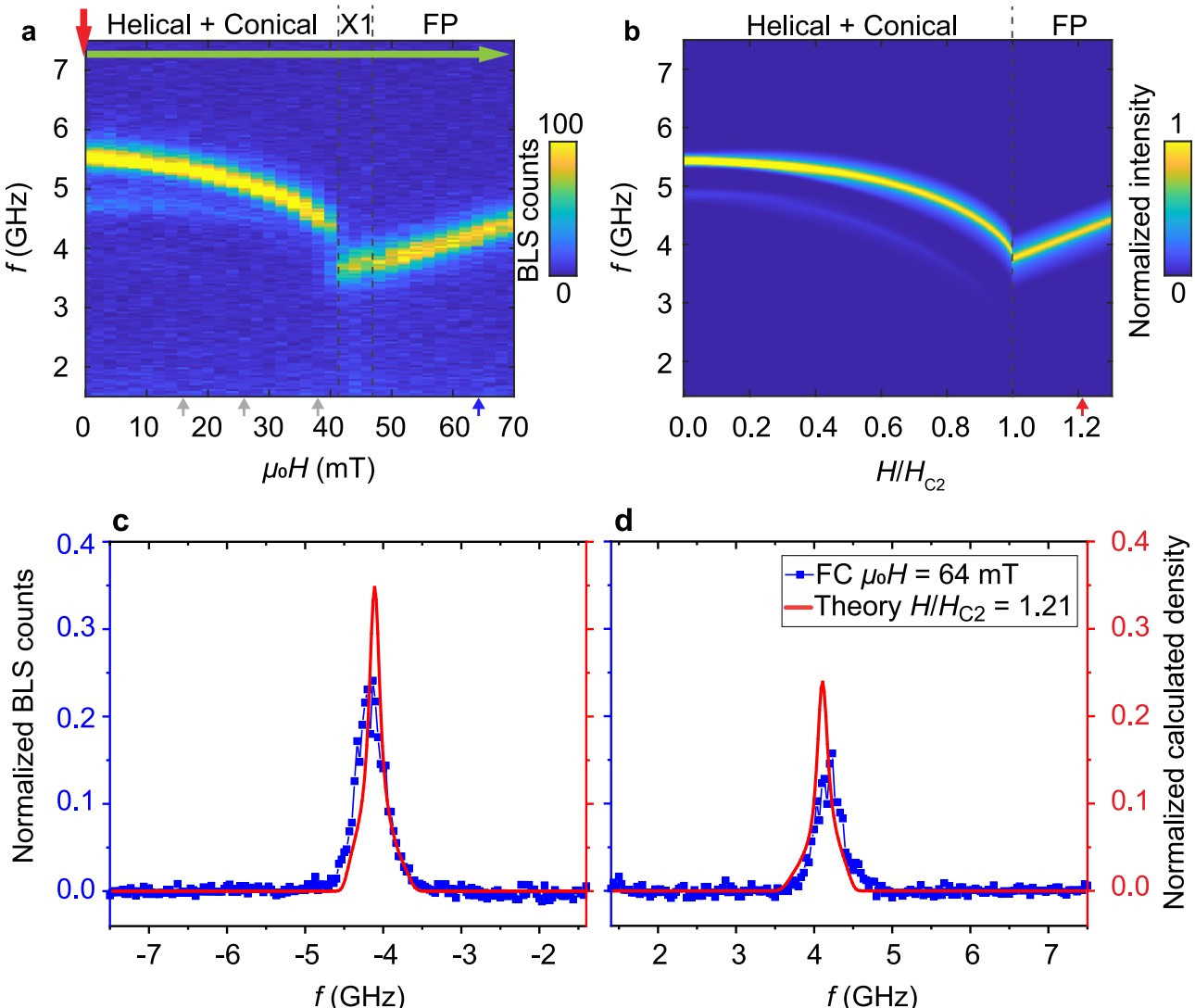

**Fig. 7 | Experimental and theoretical BLS spectra in conical and field-polarized phase. a** Anti-Stokes BLS intensities for zero-field cooling (red arrow) down to $T = 12$ K and a subsequent field scan (green arrow). Color bar represent the BLS counts. **b** Theoretical BLS spectra of the conical phase, $H < H_{c2}$, and field polarized phase, $H > H_{c2}$, with the same normalization as in (**a**). **c** (Stokes) and **d** (anti-Stokes) shows line cuts of the spectra obtained at fixed magnetic field

$\mu_0 H = 64$ mT (blue symbols) and comparison to theory with $H/H_{c2} = 1.21$ (red line). A frequency-independent background signal (dark counts) was removed from the raw BLS spectra to realize a baseline similar to the theoretical curves. Both the experimental and theoretical data have been normalized with respect to the total spectral weight integrated over the full frequency range from $-7.5$ GHz to $-1.4$ GHz and from 1.4 GHz to 7.5 GHz.

magnetic states denoted as *Mixed* in Figs. 4a and 5. Interestingly, close to this field range the theoretically predicted frequencies of the three modes, breathing, CW, and quadrupole-2, strongly bend towards smaller frequencies partially reflecting the trend also observed experimentally. These aspects will be further investigated in future work. In the focused BLS setup the scattering probability is summed over a range of solid angles for in- and outgoing wavevectors. This gives rise to spectral line shapes that depend on the dispersion of each mode. In particular, the pronounced non-reciprocity of the dispersion of the CCW mode discussed previously[29] is reflected in its broad linewidth in spectra obtained by a focused BLS setup.

## Conclusions

In the present study we applied focused BLS as an effective tool to study spin waves in the skyrmion lattice of the chiral magnet $Cu_2OSeO_3$ with high frequency resolution. Combined with the quantitative theory which models the distribution of transferred wave vectors in the scattering process, the minibands in the first BZ were explored. Our quantitative understanding of such bands substantially promotes the control and engineering of magnonic

crystals with topological magnon band structures operating at microwave frequencies. As an outlook, we note that the small laser spot, combined with separate detection of oppositely propagating magnons via the Stokes and anti-Stokes scattering processes, might be used to identify a magnon edge mode with topological properties materializing at the border of skyrmion lattice domains[14–16]. The nontrivial bands identified here further enrich unconventional computing applications[11,41–43] via wavevector-dependent spectral weights. The results underscore the essential role of cryogenic BLS in driving the experimental exploration into these directions.

## Methods
### Sample description

The $Cu_2OSeO_3$ bulk was grown by chemical vapor transport method and shaped to the dimension of about 4 mm × 3 mm × 0.5 mm. Three axes of the samples [001] × [110] × [1$\bar{1}$0] were characterized by diffraction patterns taken with a transmission electron microscope on a lamella fabricated from the same bulk $Cu_2OSeO_3$ crystal. The top surface of the samples was polished for BLS laser focusing. A bipolar magnetic field was applied along

https://doi.org/10.1038/s43246-025-00858-4 **Article**

$\hat{x}$-axis. The sample orientation with respect to the incident laser is illustrated in Fig. 1a. The phase diagram was characterized by AC susceptibility measurements and is shown in the Supplementary Fig. 2.

## Temperature-versus-field histories

Two typical temperature-versus-field histories were exploited in the BLS experiments: ZFC with field sweeps up and field cooling (FC) with field sweeps in both directions. They are sketched in Supplementary Fig. 1. The current in the magnet coils for zero magnetic field was calibrated at room temperature. Within the protocol of ZFC, the magnetic field $\mu_0 H$ was increased from zero after the cooling process. Within the protocol of FC, the cooling process was performed with a certain cooling field $\mu_0 H_{FC}$ applied, e.g., 16 mT. After reaching the relevant temperature the field was either (i) scanned up or (ii) scanned down from the cool-down field. In between each two scans (i) and (ii), the sample was heated up to at least 100 K for 10 min and cooled down again. Spectra were collected after the temperature had been stabilized for 10 min. The Stokes and Anti-Stokes spectra at each field were collected at the same time.

## Cryogenic micro-focused Brillouin light scattering setup

We use a monochromatic continuous-wave solid-state laser with a wavelength of $\lambda_{in} = 532$ nm. Incident photons were shaped by a polarizer to carry linear polarization along **y** as indicated in Fig. 1c. The scattered photons were analyzed with a six-pass Fabry-Perot interferometer TFP-2 (JRS Scientific Instruments) which contains a polarization filter only allowing photons with linear polarization along **x**. This process filters out most phonon signals, enhancing the signal-to-noise ratio for magnon detection in BLS. A Mitutoyo M Plan Apo SL 100× objective lens with NA = 0.55 (numerical aperture) was utilized for focusing the laser on the $Cu_2OSeO_3$ sample residing in a magnetooptical cryostat. The closed-cycle cryostat was equipped with magnetic field coils. The $Cu_2OSeO_3$ sample was placed on a cold finger where a slip-stick step-scanner was installed to position the sample. Microscopy images were taken by a charge-coupled device camera so that we located the laser at specific positions during the measurements. Right beneath the sample, a thermal sensor and a heater was placed. Thereby we achieved fast cooling of the sample temperature with a rate of up to 25 K min$^{-1}$. Our cool-down procedure fuls the reported quenched metastable skyrmion lattice mechanism by fast cooling through the skyrmion lattice pocket in the phase diagram[29-32]. In our experiments, the focused BLS green laser of 0.8 mW was applied to the sample during the fast cooling process[44]. Resonances of the metastable skyrmion lattice were observed over a wide temperature range from about 10 K to 40 K (Supplementary Fig. 5). BLS spectra similar to the reported ones were recorded when we applied different cooling fields of 10 mT $\leq \mu_0 H_{FC} \leq 16$ mT with a cooling rate of $\geq 6$ K min$^{-1}$. The stable skyrmion lattice phase was noticed in BLS at a sample holder temperature of 50 K in that clearly shifted mode frequencies appeared between 10 mT to 16 mT (Supplementary Fig. 3). The discrepancy between the transition temperatures observed in BLS and in AC susceptibility measurements was attributed to the different mounting of the sample and the additional heating by the laser in the BLS experiment.

## Conical and field-polarized spectra in micro-focused BLS

As a benchmark, we present in Fig. 7a the intensity maps of experimental BLS spectra obtained in the conical helix phase and FP phase of $Cu_2OSeO_3$. The color-coded spectra of Fig. 7a contain branches (yellow) which follow a field dependency consistent with a previous work using an unfocused laser beam[33]. They substantiate the good quality of our chiral magnet $Cu_2OSeO_3$. These experimental spectra were taken in a ZFC protocol, i.e., the sample was cooled down at $H = 0$ as indicated by the red arrow, see Supplementary Fig. 1 for details. After stabilizing the temperature $T$ at 12 K, the external field was increased from 0 mT to 70 mT as indicated by the green arrow. In the field range from 0 mT to 38 mT, a mode with large intensity and a mode

with weaker spectral weight and lower frequency are clearly resolved that we attribute, respectively, to the $+Q$ and $-Q$ modes of the conical helix phase[13,20,33]. At around 40 mT, the slope of the frequency-versus-field dependence of the resonance reverses from negative to positive, which indicates a phase transition. This is confirmed by AC susceptibility measurements yielding a finite imaginary part in that field range, see Supplementary Fig. 2. There might exist an intermediate phase $X_1$ attributed to either a tilted-conical or a low-temperature skyrmion lattice phase; both phases are known, respectively, to be metastable and stable due to magnetocrystalline anisotropies in $Cu_2OSeO_3$ at low temperatures for the present field orientation[31,37,38]. For larger fields above 48mT, the resonance exhibits a field dependency typically attributed to the Kittel mode. This behavior indicates that the field polarized phase is reached.

For the conical helix and the FP phase in $Cu_2OSeO_3$ the BLS cross section was evaluated previously in ref. 33, and in Fig. 7b we present the theoretical results for the focused BLS setup. Figure 7b shows the calculated Anti-Stokes component corresponding to the absorption of magnons. In the conical phase, $H < H_{c2}$, the $+Q$ and $-Q$ mode with higher and lower frequency, respectively, can be clearly distinguished, whereas in the field polarized phase, $H > H_{c2}$, a single branch remains.

In Fig. 7c, d we show the frequency dependence of the experimental BLS cross section at a fixed magnetic field $\mu_0 H = 64$ mT (blue symbols) and comparison to theory with $H/H_{c2} = 1.21$ (red line). The asymmetry in intensity of Stokes and Anti-Stokes signal was used to fit the magneto-optic constants $G_{\alpha\beta}$. It is worth noticing that the theory captures well the spectral line shape that arises from the geometric averaging of in- and out-going photon wavevectors in the micro-focused BLS setup and, in particular, it accounts for the observed full-width-at-half-maximum.

## Theory for the chiral magnet

For the theoretical description of the magnetization dynamics of the chiral magnet we follow previous work[1,13,17,22,24,25,29,33] and use the free energy functional $\mathcal{F} = \mathcal{F}_0 + \mathcal{F}_{dip}$ with

$$\mathcal{F}_0 = \int d\mathbf{r} \left[ A(\nabla_i m_j)^2 + \sigma D \mathbf{m}(\nabla \times \mathbf{m}) - \mu_0 M_s \mathbf{m} \mathbf{H} + \lambda(1 - \mathbf{m}^2)^2 \right],$$

(6)

where $A$ is the exchange interaction, $D$ is the DMI, $M_s$ is the saturation magnetization, $\mu_0$ is the magnetic field constant, and $\mathbf{H}$ is the applied magnetic field. We assume a left-handed system with $\sigma = -1$. In this work, we focus on transversal spin fluctuations and use for the parameter $\lambda = \frac{D^2}{A} 10^4$, which fixes the length of the normalized magnetization vector $\mathbf{m}$ to unity up to deviations on the level of a percent. The magnetization field can then be identified with $\mathbf{M} = M_s \mathbf{m}$. The magnetic dipolar interaction reads

$$\mathcal{F}_{dip} = \int d\mathbf{r} d\mathbf{r}' \frac{\mu_0 M_s^2}{2} m_i(\mathbf{r}) \chi_{dip,ij}^{-1}(\mathbf{r} - \mathbf{r}') m_j(\mathbf{r}').$$

(7)

The Fourier transform of the susceptibility, $\chi_{dip,ij}^{-1}(\mathbf{q})$, for wavevectors larger than the inverse linear system size $|\mathbf{q}| \gg 1/L$ is given by $\chi_{dip,ij}^{-1}(\mathbf{q}) = \frac{q_i q_j}{\mathbf{q}^2}$ whereas in the opposite limit $|\mathbf{q}| \ll 1/L$ it can be approximated, $\chi_{dip,ij}^{-1}(0) = N_{ij}$, by the demagnetization matrix with unit trace tr$\{N_{ij}\} = 1$. In the limit of small spin-orbit coupling, corrections to this theory, which includes magnetocrystalline anisotropies, are parametrically small as they are suppressed by higher powers of spin-orbit coupling.

The magnetization dynamics in the absence of damping is described by the Landau-Lifshitz equation

$$\partial_t \mathbf{M}(\mathbf{r}, t) = \gamma_0 \mathbf{M}(\mathbf{r}, t) \times \frac{\delta \mathcal{F}}{\delta \mathbf{M}(\mathbf{r}, t)},$$

(8)

with the gyromagnetic ratio $\gamma_0 = g \mu_B / \hbar$. In order to access the magnetization dynamics, this equation was treated perturbatively in the framework of linear

spin-wave theory by expanding it up to linear order in the deviations from the static equilibrium state, $\mathbf{M}(\mathbf{r}, t) \simeq \mathbf{M}_{eq}(\mathbf{r}) + \delta\mathbf{M}(\mathbf{r}, t) = M_s(\mathbf{m}_{eq}(\mathbf{r}) + \delta\mathbf{m}(\mathbf{r}, t))$.

The above theory possesses the characteristic wavevector and frequency scale, $Q = D/(2A)$ and $\omega_{c2} = \frac{D^2\gamma_0}{2AM_s}$, respectively. The effective strength of the dipolar interaction can be quantified by the dimensionless parameter $\chi_{con}^{int} = \frac{2A\mu_0 M_s^2}{D^2}$. The mean-field ground state of the theory changes from the FP phase to the conical helix phase at the critical internal field $H_{c2}^{int} = \frac{D^2}{2A\mu_0 M_s}$, which can be also related to the frequency scale $\hbar\omega_{c2} = g\mu_0\mu_B H_{c2}^{int}$. We note that the value for the frequency scale $\omega_{c2}/2\pi = 2.03$ GHz fitted to the data in the FP phase implies for a $g$-factor of $g = 2.1$[22] an internal critical field $\mu_0 H_{c2}^{int} = 68$ mT. This overestimates the value found experimentally $\mu_0 H_{c2}^{int} = \mu_0 H_{c2}/(1 + N_x\chi_{con}^{int}) \approx 43$ mT (see Supplementary Fig. 6) where $\mu_0 H_{c2} \approx 53$ mT and the demagnetization factor $N_x \approx 0.13$. A similar discrepancy was found in the laser-based experiments performed by Ogawa et al.[33]. The origin of this inconsistency is unclear but could be due to local heating by the laser or due to the so far neglected magnetocrystalline anisotropies.

## Theory for the micro-focused BLS cross section

The light-matter interaction relevant for our purpose is given by the effective action,

$$\mathcal{S}_{int} = \int dt d\mathbf{r} \frac{1}{2}\left(\tilde{K}(\mathbf{E}\times\partial_t\mathbf{E})\cdot\mathbf{M} + G_{\mu\nu\lambda\kappa}E_\mu E_\nu M_\lambda M_\kappa\right), \quad (9)$$

where $\mathbf{E}$ is the electric field, and $\tilde{K}$ and the tensor $G_{\mu\nu\lambda\kappa}$ parametrize the magneto-optic interaction within the cubic material. The latter tensor is specified by three non-vanishing parameters only, $G_{xxxx} = G_{yyyy} = G_{zzzz} = G_{11}$, $G_{xxyy} = G_{xxzz} = G_{yyzz} = G_{12}$, and $G_{xyxy} = G_{xzxz} = G_{yzyz} = G_{44}$. Expanding the interaction in lowest order spin-wave theory, we get

$$\mathcal{S}_{int}^{(1)} = \int dt d\mathbf{r} \frac{1}{2}\left(\tilde{K}(\mathbf{E}\times\partial_t\mathbf{E})\cdot\delta\mathbf{M} + G_{\mu\nu\lambda\kappa}E_\mu E_\nu 2M_{eq,\lambda}\delta M_\kappa\right). \quad (10)$$

In order to describe the incoming light in the micro-focused BLS setup, we assume an aplanatic lens for which the focal electric field within the material can be described by the angular spectrum representation[45]

$$\mathbf{E}_{in}(\mathbf{r}) \propto \int_{\theta'_{max}} d\Omega'_{in} \mathbf{E}_{\infty,in} e^{i\mathbf{k}'_{in}\mathbf{r}}, \quad (11)$$

with the abbreviation $\int_{\theta'_{max}} d\Omega'_{in} = \int_0^{\theta'_{max}} d\theta'_{in} \sin\theta'_{in} \int_0^{2\pi} d\phi'_{in}$. The incoming wavevector $\mathbf{k}'_{in} = |\mathbf{k}'_{in}|(\mathbf{x}\sin\theta'_{in}\cos\phi'_{in} + \mathbf{y}\sin\theta'_{in}\sin\phi'_{in} - \mathbf{z}\cos\theta'_{in})$ and the amplitude distribution is given by

$$\mathbf{E}_{\infty,in} = \mathcal{E}_{in}(\theta'_{in})\sqrt{\cos\theta'_{in}/n}\left(t_s(\theta'_{in})\cos\phi'_{in}\mathbf{h}'_{1,in} + t_p(\theta'_{in})\sin\phi'_{in}\mathbf{h}'_{2,in}\right), \quad (12)$$

where $\mathbf{h}'_{1,in} = -\mathbf{x}\sin\phi'_{in} + \mathbf{y}\cos\phi'_{in}$ and $\mathbf{h}'_{2,in} = \mathbf{x}\cos\theta'_{in}\cos\phi'_{in} + \mathbf{y}\cos\theta'_{in}\sin\phi'_{in} + \mathbf{z}\sin\theta'_{in}$. These latter two vectors describe the $s$- and $p$-polarized components of the focused beam. The function $\mathcal{E}_{in}$ specifies the cylindrically symmetric beam shape, and the Fresnel amplitudes $t_{s,p}$ account for the transmission of the light through the surface of the material. The form of $\mathbf{E}_{\infty,in}$ takes into account that the polarization of the incoming beam is polarized in $\mathbf{y}$-direction. As the polar angle is relatively small $0 < \theta'_{in} < \theta'_{max}$ with $\theta'_{max} \approx 16°$, we will approximate the Fresnel amplitudes in the following by their $\theta'_{in} = 0$ limit, for which $t_s(0) = t_p(0) = t$. We will also approximate $\mathcal{E}_{in}(\theta'_{in}) \approx \mathcal{E}_{in}(0)$ such that $\mathbf{E}_{\infty,in} = \mathcal{E}_{in}(0)\sqrt{\cos\theta'_{in}/n} t \mathbf{e}'_{in}$ where the polarization vector is defined by

$$\mathbf{e}'_{in} = \cos\phi'_{in}\mathbf{h}'_{1,in} + \sin\phi'_{in}\mathbf{h}'_{2,in}. \quad (13)$$

The BLS scattering amplitude deriving from Eq. (10) for the incoming focused light beam scattering into a plane wave $\mathbf{k}'_{out}$ with frequency $\omega_{out}$ is

then proportional to

$$\int dt d\mathbf{r}\int_{\theta'_{max}} d\Omega'_{in}\sqrt{\cos\theta'_{in}}e^{i(\mathbf{k}'_{in}-\mathbf{k}'_{out})\mathbf{r}}e^{-i(\omega_{in}-\omega_{out})t}e'^*_{out,\mu}\delta\varepsilon_{\mu\nu}(\mathbf{r}, t)e'_{in,\nu}. \quad (14)$$

where we used the fluctuating part of the dielectric permittivity already introduced in Eq. (5) with $K_{\mu\nu} = K\epsilon_{\mu\nu\lambda}$ where $\epsilon_{\mu\nu\lambda}$ is the antisymmetric Levi-Civita tensor. Note that the time derivative in the effective action leads to a complex-valued permittivity that depends on the center-of-mass frequency of the in- and out-going light which is absorbed in the coefficient $K = -i\tilde{K}(\omega_{out} + \omega_{in})/2$. We also used that the polarization of the out-going light must be such that it passes the polarization filter in $\mathbf{x}$-direction of the detector,

$$\mathbf{e}'_{out} = -\sin\phi'_{out}\mathbf{h}'_{1,out} + \cos\phi'_{out}\mathbf{h}'_{2,out}, \quad (15)$$

where $\mathbf{h}'_{1/2,out}$ are similarly defined as $\mathbf{h}'_{1/2,in}$.

Using Fermi's Golden rule and thermally averaging over the initial states and summing over the final states of the magnetic subsystem we obtain the transition probability

$$P \propto \int dt\, e^{i(\omega_{in}-\omega_{out})t}\int d\mathbf{r}_1 d\mathbf{r}_2\int_{\theta'_{max}} d\Omega'_{in,1}d\Omega'_{in,2}\sqrt{\cos\theta'_{in,1}\cos\theta'_{in,2}}$$
$$\times e^{i(\mathbf{k}'_{in,1}-\mathbf{k}'_{out})\mathbf{r}_1}e^{-i(\mathbf{k}'_{in,2}-\mathbf{k}'_{out})\mathbf{r}_2}e'_{out,\mu}e'^*_{in,2,\nu}e'^*_{out,\rho}e'_{in,1,\delta}\langle\delta\varepsilon^*_{\mu\nu}(\mathbf{r}_2, t)\delta\varepsilon_{\rho\delta}(\mathbf{r}_1, 0)\rangle, \quad (16)$$

where the brackets $\langle.\rangle$ denote the thermal averaging. The incoming polarization vectors $\mathbf{e}'_{in,1}$ and $\mathbf{e}'_{in,2}$ are understood to be parametrized with angles related to the two integrations over solid angles $d\Omega'_{in,1}$ and $d\Omega'_{in,2}$. Now, consider the two spatial integrals

$$\int d\mathbf{r}_1 d\mathbf{r}_2\, e^{i(\mathbf{k}'_{in,1}-\mathbf{k}'_{out})\mathbf{r}_1}e^{-i(\mathbf{k}'_{in,2}-\mathbf{k}'_{out})\mathbf{r}_2}\langle\delta\varepsilon^*_{\mu\nu}(\mathbf{r}_2, t)\delta\varepsilon_{\rho\delta}(\mathbf{r}_1, 0)\rangle =$$
$$= \int d\mathbf{r}e^{i(\mathbf{k}'_{out}-\frac{\mathbf{k}'_{in,1}+\mathbf{k}'_{in,2}}{2})\mathbf{r}}\int d\mathbf{R}\, e^{i(\mathbf{k}'_{in,1}-\mathbf{k}'_{in,2})\mathbf{R}}\langle\delta\varepsilon^*_{\mu\nu}(\mathbf{r}_2, t)\delta\varepsilon_{\rho\delta}(\mathbf{r}_1, 0)\rangle, \quad (17)$$

that can be decomposed into an integral over the distance $\mathbf{r} = \mathbf{r}_2 - \mathbf{r}_1$ and the center-of-mass coordinate, $\mathbf{R} = (\mathbf{r}_1 + \mathbf{r}_2)/2$. For a translationally invariant system the correlation function $\langle\delta\varepsilon^*_{\mu\nu}(\mathbf{r}_2, t)\delta\varepsilon_{\rho\delta}(\mathbf{r}_1, 0)\rangle$ would be independent of $\mathbf{R}$ and its integral would fix the two incoming wavevectors to be identical, i.e., $\mathbf{k}'_{in,1} = \mathbf{k}'_{in,2}$. In the present case, the correlation function is determined by the spin-wave fluctuations of the skyrmion crystal. The skyrmion crystal breaks translational invariance such that the correlation function in fact depends on the center-of-mass coordinate $\mathbf{R}$. It depends periodically on $\mathbf{R}$ with Fourier components given by the reciprocal lattice of the skyrmion crystal. However, the spread of the incoming wavevectors within the focused beam is at most $|\mathbf{k}'_{in,1} - \mathbf{k}'_{in,2}| \leq \sqrt{2}n|\mathbf{k}_{in}|\sqrt{1 - \cos(2\theta'_{max})} \approx 12.9$ rad $\mu m^{-1}$ that is much smaller than the lowest reciprocal lattice vector of the skyrmion crystal, $k_{SkL} \approx 100$ rad $\mu m^{-1}$. Consequently, the spatial integral over $\mathbf{R}$ can only pick up the zero-wavevector component of the correlation function similar to a translationally invariant system. The transition probability thus reduces to

$$P \propto \int_{\theta'_{max}} d\Omega'_{in}\cos\theta'_{in}e'_{out,\mu}e'^*_{in,\nu}e'^*_{out,\rho}e'_{in,\delta}\langle\delta\varepsilon^*_{\mu\nu}(\mathbf{r}_2, t)\delta\varepsilon_{\rho\delta}(\mathbf{r}_1, 0)\rangle_{\mathbf{q},\omega}, \quad (18)$$

where the wavevector and frequency transferred from the light beam to the sample are $\mathbf{q} = \mathbf{k}'_{in} - \mathbf{k}'_{out}$ and $\omega = \omega_{in} - \omega_{out}$, respectively. We also abbreviated

$$\langle\delta\varepsilon^*_{\mu\nu}(\mathbf{r}_2, t)\delta\varepsilon_{\rho\delta}(\mathbf{r}_1, 0)\rangle_{\mathbf{q},\omega} = \int dt\, e^{i\omega t}\frac{1}{V}\int d\mathbf{r}_1 d\mathbf{r}_2 e^{-i\mathbf{q}(\mathbf{r}_2-\mathbf{r}_1)}\langle\delta\varepsilon^*_{\mu\nu}(\mathbf{r}_2, t)\delta\varepsilon_{\rho\delta}(\mathbf{r}_1, 0)\rangle, \quad (19)$$

where $V$ is the volume.

The transition probability effectively adds intensities of individual scattering processes from a wavevector $\mathbf{k}'_{in}$ within the focused beam to $\mathbf{k}'_{out}$. Interference between amplitudes of distinct $\mathbf{k}'_{in}$ does not occur as the spread of incoming wavevectors within the focused beam is too little to detect the periodicity of the skyrmion lattice. Finally, summing over outgoing wavevectors one arrives at the BLS scattering cross section of Eq. (4).

The correlation function of the dielectric permittivity in Eq. (3) can be expressed in terms of the spin correlation function using Eq. (5),

$$
\begin{aligned}
\langle \delta\varepsilon^*_{\mu\nu}(\mathbf{r}, t)\delta\varepsilon_{\rho\delta}(\mathbf{r}', t')\rangle &\approx \left( K^*_{\mu\nu\xi} + 2G^*_{\mu\nu\lambda\xi}M_{eq,\lambda}(\mathbf{r}) \right) \\
&\times \left( K_{\rho\delta\xi'} + 2G_{\rho\delta\lambda'\xi'}M_{eq,\lambda'}(\mathbf{r}') \right) \langle \delta M_\xi(\mathbf{r}, t)\delta M_{\xi'}(\mathbf{r}', t')\rangle.
\end{aligned}
\tag{20}
$$

The parameter $K$ of $K_{\mu\nu\lambda} = K\epsilon_{\mu\nu\lambda}$ determines the total spectral weight and can be absorbed in the overall proportionality constant of the total cross section (4). The higher-order tensor $G_{\mu\nu\delta\lambda}$ is responsible for the asymmetry between the intensities of Stokes and anti-Stokes signals. Fitting the excitation spectra within the FP phase using the theory of ref. 33 we found a reasonable agreement for the values $2iM_sG_{44}/K = 0.123$ and $G_{11} = G_{12} = 0$, which we also used for the calculation within the skyrmion lattice phase.

With the help of the fluctuation-dissipation theorem, the Fourier transform of the magnetic correlation function with respect to the time difference $t - t'$ can be related to the response function

$$
\langle \delta M_\xi(\mathbf{r}, t)\delta M_{\xi'}(\mathbf{r}', t')\rangle_\omega = \frac{2\hbar}{1 - e^{-\frac{\hbar\omega}{k_BT}}} \chi''_{\xi\xi'}(\mathbf{r}, \mathbf{r}'; \omega) \simeq \frac{2k_BT}{\omega} \chi''_{\xi\xi'}(\mathbf{r}, \mathbf{r}'; \omega),
\tag{21}
$$

with Boltzmann constant $k_B$ and temperature $T$. In the last approximation we used that we work in the limit $\hbar\omega \ll k_BT$. The correlation function that eventually enters the differential cross section (3) can thus be expressed as

$$
\begin{aligned}
\langle \delta\varepsilon^*_{\mu\nu}(\mathbf{r}, t)\delta\varepsilon_{\rho\delta}(\mathbf{r}', t')\rangle_{\mathbf{q},\omega} &= \frac{2k_BT}{\omega}\frac{1}{V}\int d\mathbf{r}d\mathbf{r}'\, e^{-i\mathbf{q}\cdot(\mathbf{r}-\mathbf{r}')}\left( K^*_{\mu\nu\xi} + 2G^*_{\mu\nu\lambda\xi}M_{eq,\lambda}(\mathbf{r}) \right) \\
&\times \left( K_{\rho\delta\xi'} + 2G_{\rho\delta\lambda'\xi'}M_{eq,\lambda'}(\mathbf{r}') \right)\chi''_{\xi\xi'}(\mathbf{r}, \mathbf{r}'; \omega).
\end{aligned}
\tag{22}
$$

For the skyrmion lattice, the equilibrium magnetization $\mathbf{M}_{eq}$ and the response function $\chi''_{\xi\xi'}$ in particular were evaluated previously in the context of inelastic neutron scattering and for details we refer the reader to the Supplementary information of ref. 17. In the present context it is important to note that the length of the reciprocal lattice vector $k_{SkL}$ depends on the magnetic field, see Supplementary Fig. 7. The time and spatial evolution of certain magnon modes is illustrated in Supplementary Fig. 8 as well as in the supplementary Movies.

Practically, it is convenient to rewrite the scattering cross section in the form

$$
\sigma(\omega) \propto \int d\mathbf{q}\, F_{\mu\nu\rho\delta}(\mathbf{q})\langle \delta\varepsilon^*_{\mu\nu}(\mathbf{r}, t)\delta\varepsilon_{\rho\delta}(\mathbf{r}', 0)\rangle_{\mathbf{q},\omega}
\tag{23}
$$

with the auxiliary tensor function

$$
\begin{aligned}
F_{\mu\nu\rho\delta}(\mathbf{q}) &= \left( \int_0^{2\pi} d\phi'_{in} \int_0^{\theta'_{max}} d\theta'_{in}\sin\theta'_{in}\cos\theta'_{in}\, e'^*_{in,\nu}e'_{in,\delta} \right) \\
&\times \left( \int_0^{2\pi} d\phi'_{out} \int_{\pi-\theta'_{max}}^{\pi} d\theta'_{out}\sin\theta'_{out}e'_{out,\mu}e'^*_{out,\rho} \right)\delta(\mathbf{q} - (\mathbf{k}'_{in} - \mathbf{k}'_{out})),
\end{aligned}
\tag{24}
$$

where the two integrals are convoluted via the delta function. This auxiliary function was determined numerically for a sufficiently dense mesh of wavevectors $\mathbf{q}$ and, afterwards, the total cross section was evaluated using Eq. (23) by discretizing the $\mathbf{q}$-integral. In addition to Figs. 4b and 6, the resulting frequency dependence of the total scattering cross section is illustrated for various magnetic fields in Supplementary Fig. 9.

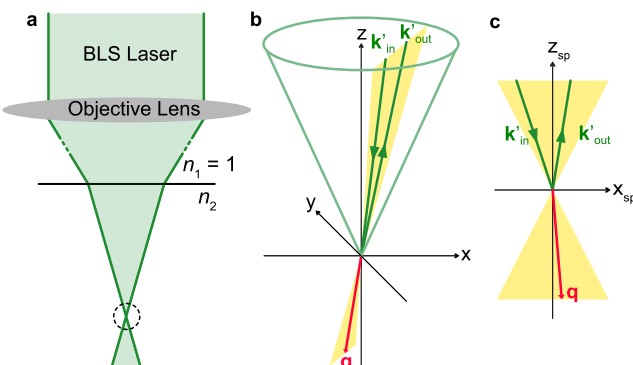

**Fig. 8 | Scattering geometry of the micro-focused BLS experimental configuration. a** Sketch of the micro-focused setup. The laser is focused on the sample top surface by the objective lens, and it is additionally bended when entering the material, where the refractive indices, $n_1 = 1$ and $n_2 = n$. **b** Example of a generic scattering event taking place in the experimental setup close to the objective lens' focal point, that is indicated by the dashed circle in (**a**). The opening angle $\theta'_{max}$ of the green cone is determined by the aperture of the lens. The wavevectors of the incoming photon, $\mathbf{k}'_{in}$, the scattered magnon, $\mathbf{q}$, and the outgoing photon, $\mathbf{k}'_{out}$, are all located within the same scattering plane indicated by the yellow surface. **c** shows the same scattering event in the corresponding scattering plane; whereas the $\mathbf{x}_{sp}$-axis is located within the $\mathbf{xy}$-plane, the $\mathbf{z}_{sp}$-axis is generally tilted away from the $\mathbf{z}$-axis of (**b**).

The range of magnon wavevectors contributing to the scattering intensity via the delta function in Eq. (24) were cited in the main text and can be understood with the help of Fig. 8b, c. The transferred magnon wavevector, $\mathbf{q}$, is given by the difference between wavevectors of incoming and outgoing photons, $\mathbf{k}'_{in}$ and $\mathbf{k}'_{out}$, which also define the corresponding scattering plane. The magnitude of the transferred wavevector can be estimated as follows

$$
|\mathbf{q}| = \sqrt{\left(\mathbf{k}'_{in} - \mathbf{k}'_{out}\right)^2} \approx \sqrt{2}n|\mathbf{k}_{in}|\sqrt{1 - \frac{\mathbf{k}'_{in}\cdot\mathbf{k}'_{out}}{|\mathbf{k}'_{in}||\mathbf{k}'_{out}|}}
\tag{25}
$$

where we neglected corrections on the order of $10^{-6}$ due to the transferred frequency $\omega$, see Eq. (2). The range of values for $|\mathbf{q}|$ depends on the refractive index within the sample $n$, the wavelength of the laser $\lambda_{in} = 2\pi/|\mathbf{k}_{in}|$ and the aperture of the objective lens, which determines the cone angle $\theta'_{max}$ of the incoming light. The maximal transferred wavevector is attained for the backscattering geometry for which $\mathbf{k}'_{out} \approx -\mathbf{k}'_{in}$ such that $|\mathbf{q}| \approx 2n|\mathbf{k}_{in}| \approx 48.0$ rad $\mu m^{-1}$. The minimal value for $|\mathbf{q}|$ is obtained if the angle enclosed between in- and outgoing photon wavevectors is minimal that is $\pi - 2\theta'_{max}$ for which $|\mathbf{q}| \approx 46.2$ rad $\mu m^{-1}$. If the scattering plane is perpendicular to the applied magnetic field, the component of the magnon wavevector $q_\parallel$ longitudinal to the field remains zero. In this case, $|\mathbf{q}| = |\mathbf{q}_\perp|$ and the component $\mathbf{q}_\perp$ perpendicular to the field covers the full range of accessible magnon wavevectors, i.e., from 46.2 to 48 rad $\mu m^{-1}$. The longitudinal component $q_\parallel$ becomes extremal if the applied field is lying within the scattering plane and the angle enclosed between the photon wavevectors is minimal. In this case $|q_\parallel| = 2n|\mathbf{k}_{in}|\sin\theta'_{max} \approx 12.9$ rad $\mu m^{-1}$. This range of magnon wavevectors define the extension of the red annulus segments in Fig. 2a, b as well as the red shaded regime in panel c.

## Data availability
The data that support the findings of this study are available from the corresponding author upon reasonable request.

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

## Acknowledgements

The authors acknowledge Bin Lu for the assistance on the cryogenic setup optimization. M.G. thanks Carsten Rockstuhl for helpful discussions. We acknowledge the financial support from Swiss National Science Foundation (SNSF) via Sinergia Network NanoSkyrmionics CRSII5 171003. M.G. acknowledges support from Deutsche Forschungsgemeinschaft (DFG) via Project-id 403030645 (SPP 2137 Skyrmionics) and Project-id 445312953.

## Author contributions

D.G. and P.C. planned the experiments. A.M. and H.B. grew the sample. P.C., P.R.B., T.S. and H.M.R. characterized the sample. P.C. performed the cryogenic BLS experiments. R.C., V.K. and M.G. conducted the theoretical calculation. P.C., R.C., M.G. and D.G. analyzed the data. P.C., R.C., M.G. and D.G. wrote the manuscript with the input from all authors.

## Competing interests

The authors declare no competing interests.
