## [Transparent Peer Review file · Communications Materials]

Short-wave magnons with multipole spin precession detected in the topological bands of a skyrmion lattice

Corresponding Author: Dr Ping Che

Version 0:

Decision Letter:

Dear Dr Che,

Thank you for submitting your manuscript, "Short-wave magnons with multipole spin precession detected in the topological bands of a skyrmion lattice", to Communications Materials. It has now been seen by 3 referees, whose comments are appended below. You will see that while they find your work of interest, some important points are raised. We are interested in the possibility of publishing your study in Communications Materials, but would like to consider your response to these concerns in the form of a revised manuscript before we make a decision on publication.

We therefore invite you to revise and resubmit your manuscript, taking into account the points raised.

When submitting your revised manuscript, please include the following:

-A response letter with a point-by-point reply to each of the referee comments and a description of changes made. Please include the complete referee report in the response letter. Please note that the response letter must be separate to the cover letter to the editors.

-A marked-up version of the manuscript with all changes to the text in a different colored font. Please do not include tracked changes or comments. Please select the file type 'Revised Manuscript - Marked Up' when uploading the manuscript file to our online system.

-A clean version of the manuscript. Please select the file type 'Article File'.

-An updated <https://www.nature.com/documents/nr-editorial-policy-checklist.zip> Editorial Policy checklist, uploaded as a 'Related Manuscript File' type. This checklist is to ensure your paper complies with all relevant editorial policies. If needed, please revise your manuscript in response to these points. Please note that this form is a dynamic 'smart pdf' and must therefore be downloaded and completed in Adobe Reader. Clicking this link will download a zip file containing the pdf.

In the event that your manuscript is accepted we will provide detailed guidance on our journal policies and formatting. You may however wish to ensure that the manuscript complies with our house style at this stage. See our style and formatting guide (<https://www.nature.com/documents/commsj-phys-style-formatting-guide-accept.pdf>) and checklist (<https://www.nature.com/documents/commsj-phys-style-formatting-checklist-article.pdf>) for reference.

Data availability statements and data citations policy: All Communications Materials manuscripts must include a section titled "Data Availability" at the end of the Methods section or main text (if no Methods). More information on this policy, and a list of examples, is available at <http://www.nature.com/authors/policies/data/data-availability-statements-data-citations.pdf>.

- Accession codes for deposited data
- Other unique identifiers (such as DOIs and hyperlinks for any other datasets)
- At a minimum, a statement confirming that all relevant data are available from the authors
- If applicable, a statement regarding data available with restrictions
- If a dataset has a Digital Object Identifier (DOI) as its unique identifier, we strongly encourage including this in the Reference list and citing the dataset in the Data Availability Statement.

DATA SOURCES: We strongly encourage authors to deposit all new data associated with the paper in a persistent repository where they can be freely and enduringly accessed. We recommend submitting the data to discipline-specific, community-recognized repositories, where possible and a list of recommended repositories is provided at <http://www.nature.com/sdata/policies/repositories>.

If a community resource is unavailable, data can be submitted to generalist repositories such as <https://figshare.com/> or <http://datadryad.org/> Dryad Digital Repository. Please provide a unique identifier for the data (for example a DOI or a permanent URL) in the data availability statement, if possible. If the repository does not provide identifiers, we encourage authors to supply the search terms that will return the data. For data that have been obtained from publically available sources, please provide a URL and the specific data product name in the data availability statement. Data with a DOI should be further cited in the methods reference section.

Please use the following link to submit your documents:

Link Redacted

We hope to receive your revised paper within three months; please let us know if you aren't able to submit it within this time so that we can discuss how best to proceed. If we don't hear from you, and the revision process takes significantly longer, we will close your file. In this event, we will still be happy to reconsider your paper at a later date, as long as nothing similar has been accepted for publication at Communications Materials or published elsewhere in the meantime.

Please do not hesitate to contact me if you have any questions or would like to discuss these revisions further. We look forward to seeing the revised manuscript and thank you for the opportunity to review your work.

Best regards,

Oleksandr Pylypovskyi
Editorial Board Member
Communications Materials
orcid.org/0000-0002-5947-9760

Reviewers' comments:

Reviewer #1 (Remarks to the Author):

The manuscript by Che et al. presents a comprehensive study of the excitations and magnon bands in the insulator skyrmion hosting the chiral magnet Cu₂OSeO₃. By using microfocused Brillouin light scattering microscopy, they were able to resolve the magnon bands beyond the gamma point, obtaining unique information that could not be obtained in previous work using other methods. The manuscript thus provides novel results that are of high quality and important for understanding skyrmion motion and excitations. I therefore recommend publication in Communications Materials provided the authors consider the following points:

The text should also mention the characteristic distance between skyrmions and the wavelength of the helices.

The results Fig. 2, 3 show theoretical results and this should be mentioned in the title of the captions as well as in the text, e.g. in lines 135 and 154.

The model does not take into account anisotropy, although it is not negligible in this material, since the phase diagram depends strongly on the direction of the magnetic field with respect to the easy/hard crystallographic directions. In particular, the anisotropy becomes more pronounced at low temperatures (see e.g. Moody et al. Phys. Rev. Res. 2021, Baral et al. Phys. Rev. Res. 2021 2023, Crisanti et al. Phys. Rev. Res. 2023), which could explain the discussed discrepancy between the experimental and calculated values of $\mu_0 H_{C2}$. It is therefore not clear to what extent the anisotropy affects the outcome of the model, a point that should be included in the discussion of the results.

Reviewer #2 (Remarks to the Author):

Due to interaction of magnons with the skyrmion lattice their dispersion (dependence of their frequency on the wavevector) changes. Instead of a continuous dispersion one obtains several quantized modes, with their frequencies showing however weak wavevector dependencies. The paper presents a study of the magnon modes in a medium, where skyrmion a lattice is formed. Using Brillouin light scattering technique the authors observed magnon modes with non-zero wavevectors. This is an important step in understanding of magnon properties in media with skyrmion lattices. The experimental study is combined with a theoretical analysis, which allows identification of the observed modes. I find the paper interesting and timely. However, before publication the paper should be revised, as indicated below. It is particular important to improve the presentation of the results. In its current version the paper is very difficult to read.

1. In Figs. 1b and 3 the skyrmion lattice is illustrated. However, the scale of the 2D-picture should be shown, which allows to the reader to compare it with the wavelength of the applied light and the diameter of the focus spot.

2. Describing the light scattering process, the authors write: Compared to the wavevectors of interest, the Faraday rotation (35) is small and will be neglected. How can one the 'Faraday rotation with the wavevector'?

3. In the text the authors use notation q_{par} and q_{perp} , whereas in Fig 2 one sees q_{100} , q_{010} , and q_{001} . How these values are connected?

4. It is also not clear for what wavevector data presented in Fig. 4 correspond?

5. I have a problem with Eqs. 4 and 5. Eq. 4 represents the scattering cross-section, i.e., the scattering intensity. And integration of this intensity over the scattering angle (Eq.5) should give the result, representing the measured signal.

However, as far as I understand each mode can contribute to scattering at different angles since the mode profile is not a plane sinusoidal function. Thus, the light waves/beams scattered by a given mode to different angles are coherent to each other. For this reason, to reproduce the experimental signal, one needs to integrate the fields of the scattered light and after that to build the total intensity.

Finally, I suggest publication after the authors revise the manuscript according to my points of criticism.

Reviewer #3 (Remarks to the Author):

The reviewed work is dedicated to studying eigenexcitations of the skyrmion lattice of the noncentrosymmetric chiral magnet Cu_2OSeO_3 by microfocused Brillouin light scattering. The main achievement of the work is the registration of multipole magnon modes (the so-called quadrupole-2 and sextupole-2 modes), which were not previously available for experimental observation by other methods. The observations were performed when the wavevector length of the detected multipole spin waves is proportional to the reciprocal lattice vector of the periodic skyrmion texture, a condition that is very suitable for using such periodic structures as self-ordered magnon crystals.

I believe that this work opens the way for further detailed study of the nonreciprocal dynamics of topological magnon bands in a magnetic skyrmion lattice and their possible application to adaptive reservoir computing and thus will be of interest to a wide range of scientists working in the field of magnetism, in particular magnetics and spintronics.

The work is an original study performed at a high experimental and theoretical level. It is clearly written and nicely presented. The conclusions are well substantiated and confirmed by the agreement between the experimental observations and theoretical analysis.

The data provided are sufficient to allow independent reproduction of the results.

It is worth noting that the attached videos help to understand the dynamics of various excitations in this rather complex magnetic system.

I can recommend this manuscript for publication in its current form, except for a few recommendations.

It seems to me that the authors would need to explain to the reader more clearly, and possibly with the addition of vector scattering diagrams, the fact that the method used is sensitive to spin waves with wave vectors along the field in the range of $\pm 12.9 \text{ rad}/\mu\text{m}$ and across the field from $46.2 \text{ rad}/\mu\text{m}$ to $48 \text{ rad}/\mu\text{m}$.

Of course, this can be calculated independently in the backscattering paradigm and using the experimental setup data provided by the authors. Still, given the importance of this information for this work, it is desirable to make it as accessible as possible to non-experts.

Also, the caption to Fig. 1 states, "The light green arrow represents a photon that is scattered after emitting a magnon (red arrow) in the sample and detected with a polarization filter e_{out} ." If I'm not mistaken, this is the only place where the polarization filter is mentioned in the paper. It would be nice if the authors could describe its role in more detail. Is it limited to the need to select magnon and phonon signals, or does it also affect the selection of magnon modes?

** Visit Nature Research's author and referees' website at www.nature.com/authors for information about policies, services and author

benefits**

Communications Materials is committed to improving transparency in authorship. As part of our efforts in this direction, we are now requesting that all authors identified as 'corresponding author' create and link their Open Researcher and Contributor Identifier (ORCID) with their account on the Manuscript Tracking System prior to acceptance. ORCID helps the scientific community achieve unambiguous attribution of all scholarly contributions. You can create and link your ORCID from the home page of the Manuscript Tracking System by clicking on 'Modify my Springer Nature account' and following the instructions in the link below. Please also inform all co-authors that they can add their ORCID to their accounts and that they must do so prior to acceptance.

Version 1:

Decision Letter:

Dear Dr Che,

Your manuscript titled "Short-wave magnons with multipole spin precession detected in the topological bands of a skyrmion lattice" has now been seen again by our referees, whose comments appear below. In light of their advice I am delighted to say that we are happy, in principle, to publish a suitably revised version in Communications Materials.

We therefore invite you to edit your manuscript to comply with our journal policies and formatting style in order to maximise the accessibility and therefore the impact of your work.

EDITORIAL REQUESTS

* Your manuscript should comply with our policies and format requirements, detailed in our style and formatting guide (<https://www.nature.com/documents/commsj-phys-style-formatting-guide-accept.pdf>).

* Please edit your manuscript according to the editorial requests in the attached table, and outline revisions made in the right hand column. If you have any questions or concerns about any of our requests, please do not hesitate to contact me. It is important that each request be addressed in order to avoid delays in accepting your manuscript. Please upload the completed table with your manuscript files as a Related Manuscript file.

* The editorial requests table also includes a full list of the files that must be provided upon resubmission. Please upload your files according to this table.

* An updated editorial policy checklist that verifies compliance with all required editorial policies must be completed and uploaded with the revised manuscript. All points on the policy checklist must be addressed; if needed, please revise your manuscript in response to these points. Please note that this form is a dynamic 'smart pdf' and must therefore be downloaded and completed in Adobe Reader. Clicking this link will download a zip file containing the pdf.

OPEN ACCESS

Communications Materials is a fully open access journal. Articles are made freely accessible on publication. For further information about article processing charges, open access funding, and advice and support from Nature Research, please visit <https://www.nature.com/commsmat/open-access>

Please use the following link to submit your revised files:

Link Redacted

We hope to hear from you within two weeks; please let us know if the process may take longer.

Best regards,

Oleksandr Pylypovskyi
Editorial Board Member
Communications Materials
orcid.org/0000-0002-5947-9760

REVIEWERS' COMMENTS:

Reviewer #1 (Remarks to the Author):

The authors have addressed my concerns and the manuscript can be accepted for publication in its present form.

Reviewer #2 (Remarks to the Author):

The authors have properly responded to my points of criticism and have essentially improved the quality of the manuscript.

I recommend the current version of the paper for publication.

Reviewer #3 (Remarks to the Author):

I believe that the authors have fully satisfied my recommendations for improving the presentation of their work and think that, given my preliminary assessment of the quality of this work, the manuscript can be published in its current form.

Detailed response to reviewers' reports:

Reviewers' comments are in blue color in the following. Our response is in black color.

Reviewer #1 (Remarks to the Author): The manuscript by Che et al. presents a comprehensive study of the excitations and magnon bands in the insulator skyrmion hosting the chiral magnet Cu_2OSeO_3 . By using microfocused Brillouin light scattering microscopy, they were able to resolve the magnon bands beyond the gamma point, obtaining unique information that could not be obtained in previous work using other methods. The manuscript thus provides novel results that are of high quality and important for understanding skyrmion motion and excitations. I therefore recommend publication in Communications Materials provided the authors consider the following points:

Response: We thank Reviewer #1 for the positive comments and the recommendation to publish it in Communications Materials after revision. Below, we provide detailed responses to the points raised.

The text should also mention the characteristic distance between skyrmions and the wavelength of the helices.

Response: We have followed this recommendation and added the characteristic distance of $a_{\text{SKL}} \approx 72 \text{ nm}$ to Fig. 1b, Fig. 3 and the main text. We have also explained the weak field dependence of this parameter, which is reported in Fig. S7 of the supplementary information.

The results Fig. 2, 3 show theoretical results and this should be mentioned in the title of the captions as well as in the text, e.g. in lines 135 and 154.

Response: We have added the specification in the figure captions and the main text.

The model does not take into account anisotropy, although it is not negligible in this material, since the phase diagram depends strongly on the direction of the magnetic field with respect to the easy/hard crystallographic directions. In particular, the anisotropy becomes more pronounced at low temperatures (see e.g. Moody et al. Phys. Rev. Res. 2021, Baral et al. Phys. Rev. Res. 2021 2023, Crisanti et al. Phys. Rev. Res. 2023), which could explain the discussed discrepancy between the experimental and calculated values of $\mu_0 H_{c2}$. It is therefore not clear to what extent the anisotropy affects the outcome of the model, a point that should be included in the discussion of the results.

Response: We agree with the referee that magnetocrystalline anisotropies are potentially important. We also would like to point out that this issue is already transparently discussed in the previous version of our manuscript. In particular, below Eq. (6) we discuss the importance of anisotropies for the low-temperature phase diagram. In addition, a whole paragraph in the section "Discussion" is devoted to this issue, see "Corrections beyond the standard theory ...". Moreover, in the section "Methods" we write in the context of the fitted value for the critical field: "The origin of this inconsistency is unclear but could be due to local heating by the laser or due to the so far neglected magnetocrystalline anisotropies." As noted in the section "Discussion", the reconstruction of the experimental spectra at low fields might in fact indicate a phase with an elongated skyrmion lattice that is an excited prospect for a future study. We thank the referee for the given references that we will include in our future work where we will address the issue of magnetocrystalline anisotropies in

more detail.

Reviewer #2 (Remarks to the Author): Due to interaction of magnons with the skyrmion lattice their dispersion (dependence of their frequency on the wavevector) changes. Instead of a continuous dispersion one obtains several quantized modes, with their frequencies showing however weak wavevector dependencies. The paper presents a study of the magnon modes in a medium, where skyrmion a lattice is formed. Using Brillouin light scattering technique the authors observed magnon modes with non-zero wavevectors. This is an important step in understanding of magnon properties in media with skyrmion lattices. The experimental study is combined with a theoretical analysis, which allows identification of the observed modes. I find the paper interesting and timely. However, before publication the paper should be revised, as indicated below. It is particular important to improve the presentation of the results. In its current version the paper is very difficult to read.

Response: We thank Reviewer #2 for the appreciation of our work and for the suggestions which helped us to improve our manuscript. We have revised the abstract and clarified key points in response to the reviewer’s comments below to improve clarity and readability. Below we reply to the reviewer’s comments in detail.

1. In Figs. 1b and 3 the skyrmion lattice is illustrated. However, the scale of the 2D-picture should be shown, which allows to the reader to compare it with the wavelength of the applied light and the diameter of the focus spot.

Response: We have followed this recommendation and added the scale in Fig. 1b and Fig. 3.

2. Describing the light scattering process, the authors write: Compared to the wavevectors of interest, the Faraday rotation (35) is small and will be neglected. How can one the ‘Faraday rotation with the wavevector?’

Response: The Faraday effect consists of a magneto-optical phenomenon that is caused by the light propagating at different speeds depending on their polarisation. It is usually detected by measuring the polarisation rotation per unit length that a circularly polarised wave acquires during its propagation. This effect has already been measured in Cu_2OSeO_3 as a function of temperature and applied field by Versteeg et al. At a temperature of 15 K they observed a maximum polarisation rotation up to $\theta_F = 0.003$ rad per μm . If we compare this inverse length-scale with the typical momentum transfer considered in our experimental setup, it is four orders of magnitude smaller. We added a short discussion regarding the Farady effect in the main text to better explain the comparison.

3. In the text the authors use notation q_{\parallel} and q_{\perp} , whereas in Fig 2 one sees q_{100} , q_{010} , and q_{001} . How these values are connected?

Response: We thank Reviewer #2 for pointing out a possible source of confusion. The external magnetic field points along the crystallographic direction $[001] \parallel \hat{\mathbf{x}}$, as indicated by Fig. 1 and written in the main text. This determines the parallel component of the magnon wavevector, $\mathbf{q}_{\parallel} = \hat{\mathbf{x}}q_{\parallel}$, as well as the orthogonal component by $q_{\perp} = \sqrt{|\mathbf{q}|^2 - q_{\parallel}^2}$, with direction depending on the specific scattering wavevectors \mathbf{k}_{in} and \mathbf{k}_{out} . It is worth noticing that we used the notation \mathbf{q}_{\parallel} and \mathbf{q}_{\perp} when referring to the transferred magnon wavevector accessible by the BLS setup. On the contrary, q_{100} ,

q_{010} , and q_{001} were used to describe a generic wavevector along the corresponding crystallographic directions. We modified the main text to better explain the difference between the two notations.

4. It is also not clear for what wavevector data presented in Fig. 4 correspond?

Response: The spectra presented in Fig. 4 is obtained from the micro-focused BLS setup shown in Fig. 1. Consequently, each magnon band results from averaging over the entire wavevector domain, which is indicated by the red annulus regions in Fig. 2. This does not correspond to a single magnon wavevector due to the presence of the focusing lens. We have added this in the Fig. 4 caption.

5. I have a problem with Eqs. 4 and 5. Eq. 4 represents the scattering cross-section, i.e., the scattering intensity. And integration of this intensity over the scattering angle (Eq.5) should give the result, representing the measured signal. However, as far as I understand each mode can contribute to scattering at different angles since the mode profile is not a plane sinusoidal function. Thus, the light waves/beams scattered by a given mode to different angles are coherent to each other. For this reason, to reproduce the experimental signal, one needs to integrate the fields of the scattered light and after that to build the total intensity.

Response: We thank the referee for this very important, enlightening question. The referee is right that in general in a micro-focused setup the scattering amplitude of the focused light beam needs to be computed first before the scattering intensity is evaluated. This is in particular the case when one probes defects, like magnetic domain walls, that break translational invariance. In our case, nevertheless, the addition of intensities instead of amplitudes is justified. The skyrmion lattice breaks translational invariance and leads to a spatial periodicity characterized by the reciprocal lattice vectors k_{SKL} . These reciprocal lattice vectors are, however, larger than the spread of the incoming photon wavevectors within the focused beam. As a consequence, interference effects are suppressed. We made an effort and explained this issue now in detail in our modified Methods section.

In carefully deriving the expression for the total BLS scattering cross section, we also noted a mistake in our formula. Previously we did not properly take into account the intensity distribution within the beam. This has now been corrected. It concerns the $\cos \theta'_{\text{in}}$ factor in Eq. (4) whose derivation is discussed in the context of Eq. (11) in the new version of our manuscript. This correction did however not affect substantially the theoretically computed spectra that remained essentially unchanged. We thank again the referee for this thoughtful comment. We believe that our modified text substantially improved the justification as well as the presentation of our results.

Finally, I suggest publication after the authors revise the manuscript according to my points of criticism.

Response: We thank Reviewer #2 for the recommendation of publication. We appreciate all the comments and hope that Reviewer 2 will be satisfied with our revisions.

Reviewer #3 (Remarks to the Author): The reviewed work is dedicated to studying eigenexcitations of the skyrmion lattice of the noncentrosymmetric chiral magnet Cu_2OSeO_3 by microfocused Brillouin light scattering. The main achievement of the work is the registration of multipole magnon modes (the so-called quadrupole-2 and sextupole-2 modes), which were not previously available for experimental observation by other methods. The observations were performed when the wavevector length of the detected multipole spin waves is proportional to the reciprocal lattice vector of the

periodic skyrmion texture, a condition that is very suitable for using such periodic structures as self-ordered magnon crystals.

I believe that this work opens the way for further detailed study of the nonreciprocal dynamics of topological magnon bands in a magnetic skyrmion lattice and their possible application to adaptive reservoir computing and thus will be of interest to a wide range of scientists working in the field of magnetism, in particular magnetics and spintronics.

The work is an original study performed at a high experimental and theoretical level. It is clearly written and nicely presented. The conclusions are well substantiated and confirmed by the agreement between the experimental observations and theoretical analysis.

The data provided are sufficient to allow independent reproduction of the results.

It is worth noting that the attached videos help to understand the dynamics of various excitations in this rather complex magnetic system.

I can recommend this manuscript for publication in its current form, except for a few recommendations.

Response: We are greatly delighted that Reviewer #3 recognizes the significance of our findings and their potential impact on the field. We further improved the manuscript by addressing the comments of Reviewer #3.

It seems to me that the authors would need to explain to the reader more clearly, and possibly with the addition of vector scattering diagrams, the fact that the method used is sensitive to spin waves with wave vectors along the field in the range of ± 12.9 rad/ μm and across the field from 46.2 rad/ μm to 48 rad/ μm . Of course, this can be calculated independently in the backscattering paradigm and using the experimental setup data provided by the authors. Still, given the importance of this information for this work, it is desirable to make it as accessible as possible to non-experts.

Response: We thank the referee for this useful suggestion. We added a new figure containing the scattering diagram (Fig. 8) at the end of the section Methods. We also explain there in detail how the range of wavevectors is obtained. We believe that this indeed improves the accessibility of the treatment.

Also, the caption to Fig. 1 states, “The light green arrow represents a photon that is scattered after emitting a magnon (red arrow) in the sample and detected with a polarization filter e_{out} .” If I’m not mistaken, this is the only place where the polarization filter is mentioned in the paper. It would be nice if the authors could describe its role in more detail. Is it limited to the need to select magnon and phonon signals, or does it also affect the selection of magnon modes?

Response: We thank reviewer for this comments and have added the function of polarization filter in the Methods “Cryogenic micro-focused Brillouin light scattering setup” section.

Response from Authors: We acknowledge all the comments given by the reviewer which have helped us to further improve the quality of the manuscript.

Brief list of changes:

- * Abstract revised for key points clarification
- * Scale and lattice constants added in Fig. 1b, Fig. 3 and the main text

- * Specification of theoretical modeling in Fig. 2 and 3 captions
- * Wavevector range added in Fig. 4 caption
- * Discussion of the Faraday rotation added in the main text
- * Polarization filter function described in the Methods section
- * Figure S7 corrected for the field dependence of the reciprocal skyrmion lattice vector
- * Figure 8 added to the method section together with a brief description to better explain the scattering geometry

Response to reviewers' reports:

Reviewers' comments are in blue color in the following. Our response is in black color.

Reviewer #1 (Remarks to the Author):

The authors have addressed my concerns and the manuscript can be accepted for publication in its present form.

Response: We thank Reviewer #1 for the positive recommendation of our manuscript for publication.

Reviewer #2 (Remarks to the Author):

The authors have properly responded to my points of criticism and have essentially improved the quality of the manuscript.

I recommend the current version of the paper for publication.

Response: We thank Reviewer #2 for the positive recommendation of our manuscript for publication.

Reviewer #3 (Remarks to the Author):

I believe that the authors have fully satisfied my recommendations for improving the presentation of their work and think that, given my preliminary assessment of the quality of this work, the manuscript can be published in its current form.

Response: We thank Reviewer #3 for the positive recommendation of our manuscript for publication.